# Helical ultrastructure of the metalloprotease meprin α in complex with a small molecule inhibitor

Charles Bayly-Jones [1,2,9], Christopher J. Lupton [1,2,9], Claudia Fritz [3,9], Hariprasad Venugopal[4], Daniel Ramsbeck [3], Michael Wermann[3], Christian Jäger [5], Alex de Marco [1,2], Stephan Schilling[3,6], Dagmar Schlenzig [3] ✉ & James C. Whisstock [1,2,7,8] ✉

The zinc-dependent metalloprotease meprin α is predominantly expressed in the brush border membrane of proximal tubules in the kidney and enterocytes in the small intestine and colon. In normal tissue homeostasis meprin α performs key roles in inflammation, immunity, and extracellular matrix remodelling. Dysregulated meprin α is associated with acute kidney injury, sepsis, urinary tract infection, metastatic colorectal carcinoma, and inflammatory bowel disease. Accordingly, meprin α is the target of drug discovery programs. In contrast to meprin β, meprin α is secreted into the extracellular space, whereupon it oligomerises to form giant assemblies and is the largest extracellular protease identified to date (~6 MDa). Here, using cryo-electron microscopy, we determine the high-resolution structure of the zymogen and mature form of meprin α, as well as the structure of the active form in complex with a prototype small molecule inhibitor and human fetuin-B. Our data reveal that meprin α forms a giant, flexible, left-handed helical assembly of roughly 22 nm in diameter. We find that oligomerisation improves proteolytic and thermal stability but does not impact substrate specificity or enzymatic activity. Furthermore, structural comparison with meprin β reveal unique features of the active site of meprin α, and helical assembly more broadly.

Proteases critically underpin numerous processes on various levels via post-translational modification of proteins; these include cellular functions such as cell proliferation and differentiation[1,2], necrosis[3] and apoptosis[4], angiogenesis[5], migration[6,7] and more. Resultantly, proteases represent a fundamentally crucial component of normal cellular function and as such abnormal expression or dysregulation can be linked to various diseases. For these reasons, proteases are the attention of drug discovery programs and are exploited as diagnostic markers[8].

Astacin proteases, a subfamily of metzincin superfamily, are found in a variety of different species of vertebrates, invertebrates, and bacteria. Meprin α, together with the evolutionary related meprin β,

[1]Biomedicine Discovery Institute, Department of Biochemistry and Molecular Biology, Monash University, Melbourne, VIC, Australia. [2]ARC Centre of Excellence in Advanced Molecular Imaging, Monash University, Melbourne, VIC, Australia. [3]Department for Drug Design and Target Validation (IZI-MWT), Fraunhofer Institute for Cell Therapy and Immunology, Halle, Germany. [4]Ramaciotti Centre for Cryo-Electron Microscopy, Monash University, Clayton 3800 VIC, Australia. [5]Vivoryon Therapeutics N. V., Halle, Germany. [6]Hochschule Anhalt, University of Applied Sciences, Köthen, Germany. [7]EMBL Australia, Monash University, Melbourne, VIC 3800, Australia. [8]ACRF Department of Cancer Biology and Therapeutics, John Curtin School of Medical Research, Australian National University, Canberra, ACT 2601, Australia. [9]These authors contributed equally: Charles Bayly-Jones, Christopher J. Lupton, Claudia Fritz. ✉e-mail: dagmar.schlenzig@izi.fraunhofer.de; james.whisstock@monash.edu

represent a subgroup of astacins[9]. As with all astacins, the primary structure of the active site is characterized by two conserved motifs: the zinc-binding motif (HExxHxxGFxHExxRxDRD) and the methionine-turn SxMHY[10,11].

Both meprin α and β are highly expressed in epithelial cells of kidney and intestine and they have been demonstrated to a minor extent in intestinal leukocytes, skin, lung, brain, and certain cancer cells[11-17]. Within these tissues, numerous potential substrates are present including bioactive proteins and peptides, growth factors, adhesion molecules, and components of the extracellular matrix (ECM)[18]. Differential regulation and localisation of meprins provides access to these substrates, enabling specific functions in pro- and anti-inflammatory responses, ECM assembly and remodelling, wound healing, cytokine activation and signalling, as well as cell-cell adhesion[19-28].

A prominent example is the pro-collagenase function. Both meprin α and β are N- and C-pro-collagenases of pro-collagen I and III and are, as such, important for collagen assembly and tensile strength[19,26,29]. Likewise, the pro-inflammatory IL1β is activated by both enzymes, although the product resulting from processing by meprin α is two-fold more active than that of meprin β[30,31]. IL6 is inactivated by meprin α and β suggesting an anti-inflammatory effect, however the soluble IL6 receptor is a product of both activities[20,32]. In contrast, pro-migratory MCP-1 and pro-inflammatory IL18 are activated by meprin α and β respectively[33,34]. These examples reflect the complexity of effects that meprins exert and participate in, with respect to pathological functions like acute kidney injury, sepsis, urinary tract infection, and inflammatory bowel disease (IBD). The roles of meprins in inflammation and modulation of immune cells was the recent topic of intensive review elsewhere[11,27].

To achieve their function, both meprins cleave a specific but distinct motif. Both meprin α and β are characterised by a striking preference for negatively charged amino acids in the P1' position unlike most other extracellular proteases[35]. Nevertheless, meprin α and β partially discriminate between substrates, resulting in distinct activity profiles. These differences are thought to drive distinct functional properties in vivo. Nevertheless, as exemplified above, some overlap in substrates also results in redundant physiological roles.

Of note, meprin α has been suggested to drive aggressive metastatic colorectal cancer[36-38]. Epithelial cells of the normal human colon secret meprin α into the lumen, whereas in a colon carcinoma cell line meprin α is secreted in a non-polarized fashion increasing substrate accessibility. Moreover, high concentrations of meprin α activity in primary tumour stroma promotes tumour progression and metastasis. This is thought to occur due to meprin α pro-migratory and pro-angiogenic activity by way of ECM remodelling and transactivation of the EGFR/MAPK signalling pathway by shedding the ligands EGF and TGFα[39]. Interestingly, inhibition of meprin α by actinonin, a naturally occurring inhibitor, was able to abrogate these effects in vitro, suggesting meprin α is a promising target for therapeutics[39]. Importantly dysregulation of meprin α can occur in contexts where meprin β retains normal regulatory and physiological functions. As such, modulation of meprin activities may have either beneficial or deleterious effects on health depending on the context. For example, meprin β shows a protective effect in IBD and promotes mucus turnover by cleavage of MUC2, preventing bacterial overgrowth in intestinal mucosa[33,40,41].

In general, specific inhibitors for meprin α and meprin β separately might be useful to treat certain diseases, such as progressive cancers, without disrupting the physiological function of the other homologue. For these reasons, specific inhibitors of meprin α have been sought to mitigate its function in disease progression and to investigate the pathophysiological role of meprins in further detail. In order to provide controlled inhibition of meprin α functions while maintaining the physiological function of meprin β, and vice versa, significant effort has been directed to develop protease specific compounds[42-46]. However, owing to high conservation between the two enzymes this has been non-trivial and thus there currently exists a trade-off between high potency and high selectivity among existing candidates.

Both meprin α and β are expressed as glycosylated zymogens. The domain architecture of both proteases comprises an N-terminal M12A (M12A clan protease) domain, a MAM (Meprin, A-5 protein, and receptor protein-tyrosine phosphatase μ) domain, a MATH (Meprin and TNF receptor-associated factor [TRAF] homology) domain, an EGF-like domain, and, lastly, a short transmembrane and cytosolic region (Supplementary Fig. 1a). Both proteases are enzymatically processed to remove the autoinhibitory N-terminal pro-domain by trypsin-like activities (for example by the fibrinolytic protease, plasmin) in order to form the active protease[47].

A key difference between meprin α and meprin β is that the former includes a furin cleavage site between the MATH and EGF-like domain. This permits furin-mediated shedding of meprin α within the secretory pathway and, subsequently, secretion into the extracellular milieu[48,49]. In contrast, meprin β remains predominantly membrane associated, with rare shedding events mediated via ADAM10/17 proteolytic cleavage of the sequence immediately preceding the transmembrane domain[16,18,50,51]. Shedding of meprin β has been observed to drive distinct substrate specificity and function suggesting localisation affects accessibility to certain substrates[52]. Furthermore, expression of meprin α is localised in the stratum basale of the epidermis, while meprin β is found in the stratum granulosum[14]. As such, some functions of meprin α and meprin β may be dependent on their localisation and tissue distribution.

In regard to quaternary structure, both enzymes form covalently (disulphide) linked homo- and heterodimers[11]. The quaternary complex is critical for the localisation and oligomerization of meprin α. Heterodimeric meprin α/β remains tethered at the plasma membrane via membrane bound meprin β[48]. Such heterodimers can further interact to form heterotetramers[52-54]. In contrast, furin-mediated release of meprin α homodimers results in the formation of meprin α oligomeric assemblies, as has been shown for recombinant rat meprin α[53]. The non-covalent interface that mediates this homo-oligomerisation was mapped by cross-linking mass spectrometry to the MAM domain[55]. In contrast, meprin β homodimers do not oligomerise when tethered to or upon shedding from the membrane[11].

In addition to low stoichiometric assemblies, the homodimer of meprin α is capable of forming large non-covalently associated soluble oligomers with a reported size of up to 6 MDa[53]. Meprin α is, therefore, the largest known secreted protease complex. The oligomers have been characterized by means of SEC, EM and light scattering to be comprised of a heterogeneous population of ring, circle, spiral, and tube-like structures[53]. While the crystal structure of meprin β has been solved in the active, inactive and inhibitor bound states[56,57], the structure of meprin α remains to be determined.

Here, we study the structural basis for meprin α oligomerisation and its inhibition by selective small molecule inhibitors. Since the heterogeneity of meprin α oligomers precludes structure determination by X-ray crystallography, we accordingly use a cryo-EM approach to determine the structure. Our data show that both the zymogen and active form of the meprin α ectodomain (i.e., the region released through furin cleavage) is capable of self-association to form a flexible, giant helical assembly. Using a single particle like approach we determine the high-resolution reconstructions of meprin α in its zymogen and active state. Additionally, we determine structures of meprin α in the presence of a prototype selective inhibitor and the native inhibitor human fetuin-B. This structure represents a valuable tool for rational drug design and provides a basis for structural comparisons between the homologous enzymes. Lastly, we propose possible mechanisms for further investigation that may implicate meprin α polymerisation as an important component in correct function and regulation.

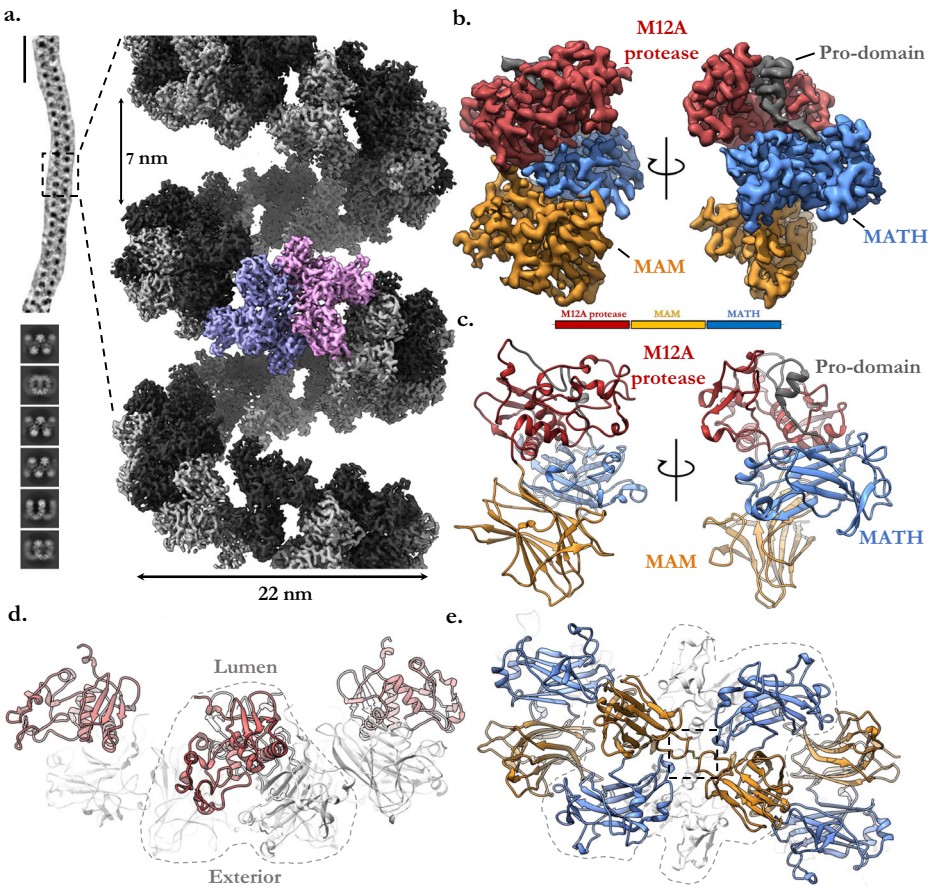

**Fig. 1 | The cryoEM reconstruction of helical meprin α. a** A representative particle of meprin α is roughly 100-150 nm in length (vertical scale bar, ~50 nm) and corresponding two-dimensional class averages of segmented meprin α particles. Dashed box shows the magnified view of the reconstructed segment. Homodimeric meprin α subunits are coloured alternating in black and grey. A central dimeric subunit is shown in purple and pink. **b** A single meprin α subunit illustrating the three globular domains (M12A [red], MAM [blue] and MATH [orange]) and the helical pro-domain (dark grey). **c** s Corresponding atomic model of a single meprin α subunit and domain schematic coloured as in **b**. **d** View along the filament axis of the meprin α helix showing a segment of four meprin α subunits. The catalytic M12A domain (red) is found on the inside of the helical tube (lumen), while the MAM and MATH domains (both light grey) are located on the periphery of the helix. **e** The meprin α helix interface is defined by two head-to-tail dimers of the MAM and MATH domains which join adjacent meprin α homodimers. In both **d** and **e** a dashed line outlines a single meprin α covalent homodimer. Dashed box contains C308 disulphide bond.

## Results

### Meprin α forms a giant flexible left-handed helical filament

It has previously been observed that meprin α function requires two proteolytic events. Firstly, the zymogen is proteolytically shed from the membrane via furin proteases, thus releasing these into the extracellular milieu. After shedding from the membrane, meprin α dimers associate non-covalently to form oligomers in the MDa range. Secondly, the zymogen is converted to the active state by proteolytic removal of the pro-peptide. To characterise the structural and biochemical properties of meprin α, we expressed a truncated recombinant construct in S2 cells that lacked both the EGF-like and transmembrane regions (Supplementary Fig. 1a, b). The secreted wild type meprin α was purified from the conditioned media via hydrophobic interaction followed by affinity chromatography. The apparent molecular mass based on SDS-PAGE was observed to be 70–73 kDa, which is notably higher than the theoretical mass of 67.7 kDa, most likely due to glycosylation (Supplementary Fig. 1c, d). Since size exclusion chromatography regularly resulted in loss of protein and activity this method was avoided. Activation of the zymogen was accomplished by way of magnetic trypsin beads. These enabled the cleavage of the pro-peptide and facilitated subsequent removal of trypsin without further dilution of the protein sample by additional purification steps.

Inspection of the recombinant material by electron microscopy revealed meprin α forms extraordinarily large and flexible supercoiled filaments (Fig. 1a, Supplementary Movie 1, Supplementary Table 1). To obtain the structure of these filaments, a cryo-ET and STA approach was initially attempted, however ultimately these reconstructions were limited by the flexibility of meprin α oligomers (Supplementary Fig. 2). Therefore, we resorted to a (pseudo) single particle analysis to overcome the intrinsic flexibility and symmetry breaking in silico (Supplementary Fig. 3; see methods for details). This approach was successful and yielded reconstructions ranging in resolution from 2.4 to 3.7 Å. The meprin α oligomer resembles a spring of roughly 22 nm in diameter and 200 nm in length (Fig. 1a, Supplementary Fig 4), with some filaments observed to be greater than 500 nm long (~27 MDa). We note that filaments of actomyosin[58] and the microtubule[59] possess extensive contacts between neighbouring turns. In contrast, the meprin α filament lacks such contacts, forming a highly spacious helical groove of roughly ~7 nm and a hollow core of ~15 nm (Fig. 1a). Resultantly, meprin α is intrinsically flexible, observed to expand, contract, and bend along the axis of the helix (Supplementary Fig. 4a; Supplementary Movie 2).

Each meprin α subunit strongly resembles the homologous protein meprin β. The subunit is a compact triad comprising the globular M12A protease, MAM and MATH domains (Fig. 1b, c). In a single

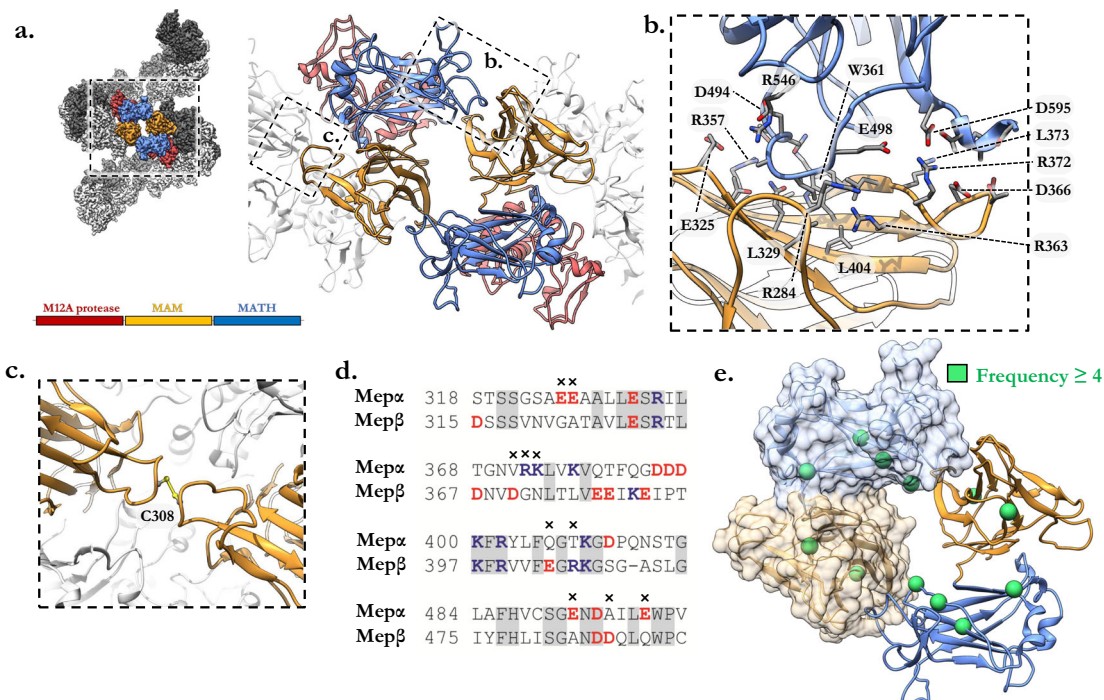

**Fig. 2 | Meprin α helices are defined by two key interfaces. a** Segment of meprin α helix showing a single helical interface (blue, orange) defined by the MAM and MATH domains. High magnification view of this region is shown (right) as a cartoon model. **b** Boxed region of the MAM/MATH interface illustrating key residue contacts forming at the helical interface. Visible is R372 which forms key salt bridges with adjacent aspartic acids. **c** Local sequence alignment of MAM/MATH domains between meprin α and meprin β showing multiple charge swap and charge loss mutations. **d** Meprin α homodimer interface showing covalent C308 disulphide bridge. **e** Frequently occurring mutation associated with cancers (curated from the COSMIC database[52] are shown as green spheres. These residues map to the MAM/MATH domains which define the helical interface.

asymmetric unit of the helix, two meprin α subunits homodimerise into a C2 arrangement via the M12A and MATH domains. The meprin α M12A domain is typical, containing the catalytic zinc ion within the active site. Each M12A domain is positioned on the inner surface of the helix, jutting inwards forming a triangular arrangement (Fig. 1d). The MAM and MATH domains buttress the protease domain, forming the outer structure of the helix interacting via a head-to-tail arrangement (Fig. 1e), while at the dimer interface a conserved disulphide bond stabilises MAM/MAM interactions (see next section). This arrangement of dimers forms an indefinite filament which coils into a left-handed helical ultrastructure.

Cisplatin treatment is known to cause an increase of meprin α in urine by inducing damage to epithelial cells in the kidney[60]. To determine whether higher-order oligomers could also be found from native source material, we conducted an analytical size-exclusion chromatography analysis of mouse urine after cisplatin treatment. Indeed, we observed evidence that native meprin α is capable of forming oligomers of similar molecular mass, albeit the major population appeared smaller than recombinant wild type meprin α (Supplementary Fig. 1e).

### Analysis of meprin α oligomerisation

The assembly of meprin α is apparently governed by two unique interfaces (Fig. 2a, b). The dimer interface is most extensive (1374 Å²) and highly conserved between both meprin α and meprin β (Supplementary Fig 5, 6; Extended Data 1). This interface is defined by the M12A protease and MAM domains, which are dependent on a conserved MAM/MAM disulphide bond to form the covalently linked homodimer (Fig. 2c). Two identical MAM/MATH interfaces (610 Å²) each provided by two homodimers together drive the head-to-tail formation of the non-covalent helical assembly of multiple homodimers (Fig. 2a, b). This latter interface is defined by extensive charge

complementarity and salt bridges between the MAM and MATH domains and has a calculated cumulative solvation free energy ($\Delta_i G$) of −5.2 kcal mol⁻¹. Consistent with this observation, cryoEM imaging of meprin α oligomers in the presence of high salt or acidic conditions resulted in disruption of the helical interface producing smaller oligomeric species (Supplementary Fig. 7). Conversely, alkaline conditions preserve the oligomeric state.

In contrast to meprin α, shed meprin β does not form higher-order oligomeric species. Superposition of MAM/MATH domains of meprin β and meprin α reveal the helical interface is abolished in meprin β due to multiple charge swap and loss-of-charge mutations (Fig. 2d). Furthermore, multiple non-conserved residues within the MAM groove result in steric effects that likely further hinder formation of a stable interface (Fig. 2b,d). In this regard, we note that frequently occurring cancer-associated mutations cluster to the MAM/MATH domains. Given that these domains define the interface of meprin α filaments, mutations that affect the fold may interfere with correct packing (Fig. 2e; Extended Data 2)[61]. In contract, meprin β cancer-associated mutations more frequently cluster within in the M12A protease domain (Extended Data 2)[61].

To assess the impact of these interfaces, we produced three mutants of meprin α by abolishing a salt bridge found at the helix interface (R372T and R372A) and the intermolecular disulphide bridge that stabilises the homodimer interface (C308A). No discernible differences could be observed between wild-type, R372T, and R372A meprin α by reducing and non-reducing SDS-PAGE, whereas mutant C308A appeared to be monomeric in all conditions (Supplementary Fig. 1c, d). MALDI-TOF mass spectrometry (MS) determined the molecular mass of meprin α C308A to be exclusively monomeric (single charged ion peak at 75 kDa), while an additional disulphide bridged dimeric species (-130 kDA) was observed for both wild-type and R372T meprin α (Supplementary Fig. 8a). This is expected, as the

absence of a disulfide bridge allows for complete dissociation of subunits in the mass spectrometer. For all variants, the zymogen form was roughly 6 kDa higher in mass than the activated counterpart consistent with the removal of the propeptide. Further, all forms had a molecular mass that exceeded the calculated mass by ~8 kDa, most likely corresponding to extensive glycosylations as observed in the cryo-EM reconstruction.

Single-molecule mass measurements by mass photometry revealed the covalent and non-covalent interfaces were disrupted by mutagenesis to different extents yielding mixed populations of different stoichiometries (Supplementary Fig. 8b). At low concentrations (10 nM) complete dissociation of the helical interface was observed for R372T, while R372A maintained weak association to assemble tetramers. Similarly, C308A maintained the ability to associate into a range of stoichiometries through the remaining non-covalent interactions, however very large oligomers were unfavoured illustrating the stabilising role of the C308 disulfide bridge. Unlike the mutant variants, very large oligomers were still observed in the single-molecule movies of wild-type material (Supplementary Fig. 8b).

Further analysis by multi angle dynamic light scattering (MADLS) revealed different hydrodynamic radii for meprin α C308A and R372T variants (Supplementary Fig. 8c). Wild type meprin α itself is polydisperse (hydrodynamic radius of main fraction ~69.6 nm), suggesting the presence of large oligomers consistent with the cryo-EM reconstruction. The observed hydrodynamic radii of C308A and R372T (~34.9 nm and ~20.8 nm respectively; Supplementary Fig. 8c) are consistent with small oligomers or dimers being formed via the remaining unaffected interface. SEC-MALS analysis of C308A and R372T gave similar predictions of molecular mass of 150 kDa and 150-175 kDa respectively (Supplementary Fig. 8d). Unlike C308A, meprin α R372T also showed some evidence of higher order oligomers under MADLS and SEC-MALS, potentially due to weak non-covalent interactions at the helix interface (Supplementary Fig. 8c,d). R372A was not analysed by SEC-MALS or mass spectrometry due to limited expression yields.

Finally, we inspected the variants C308A, R372T, R372A, and the dimeric meprin β homolog by electron microscopy. These data confirmed the occurrence of small particles of dimeric appearance for R372A and R372T, however at high concentrations (3 μM) all variants were capable of homo-oligomerising into small oligomeric species, distinct from those of wild type meprin α (Supplementary Fig. 8e). Taken together, meprin α R327T and R372A appear to be covalently linked dimers that have some low-affinity tendency to interact via the helical interface in a concentration-dependent manner. Conversely, C308A destabilises the major homodimer interface and forms elongated species via the helical interface and weakly via the homodimer interface, consistent with previous observations[55].

## Activation and substrate specificity of meprin α

Inspection of the ultrastructure reveals the active site of the protease domain is positioned symmetrically on each edge of the helical filament, with repeating sites spaced by roughly 8 nm that line the groove of the helix (Fig. 3a). Collectively, the ordered arrangement of available active sites defines a massive platform that suggests some functional purpose. We therefore sought to understand the importance of the helical assembly on meprin α substrate specificity, activity, and stability by comparing wild type meprin α to the variants, C308A, R327T and R372A.

Thus far, comparisons between the oligomeric and dimeric meprin α in vitro reveal no significant difference in specific activity nor substrate specificity. When comparing the degradation turnover of large substrates by meprin α and variant C308A no dramatic differences could be observed, except for tropoelastin degradation which appears somewhat enhanced by C308A (Supplementary Fig. 9). Similarly, the parameters for substrate hydrolysis of a small fluorogenic peptide substrate are not impacted (Fig. 3b, c). Likewise,

inhibition of meprin α by small molecule inhibitors are not impacted by the oligomeric state (Fig. 3d, e). These findings suggest allosteric effects of the oligomer do not affect function of the catalytic core. In contrast however, dimeric meprin α was found to be less stable than the oligomeric form against proteolytic degradation by trypsin and plasmin during activation of the zymogens (Fig. 3f). Furthermore, the non-oligomerising variants of meprin α showed a lower thermal stability when compared to oligomeric wild type meprin α (Fig. 3g). Therefore, oligomerisation may play a role in protein regulation by way of increasing half-life in vivo.

Unlike small molecule substrates and inhibitors, which are not sterically occluded by the helical packing, we investigated the inhibition of meprin α in vitro by murine fetuin-B, a 42 kDa globular protein. Remarkably, we observed no significant difference in the inhibitory capacity of fetuin-B between oligomeric and monomeric variants of meprin α indicating the oligomer does not interfere with binding (Fig. 3d, e). Molecular docking analysis of fetuin-B to the meprin α oligomer instead suggests a compact, intercalated packing is possible that is not impacted by the helical rise of meprin α (Supplementary Fig. 10a–d). Further, the interfaces predicted to form this meprin α/fetuin-B oligomer are conserved[62,63] (Supplementary Fig. 10f). This intercalated packing of fetuin-B is suggestive of possible higher order inhibitory oligomers. We therefore imaged meprin α in complex with human fetuin-B by cryo-EM and generated a 3.7 Å resolution reconstruction (Supplementary Fig. 10g). Indeed, this structure confirms the intricate packing suggested by the molecular docking analysis, albeit with some degree of flexibility. The presence of a conserved interface suggests the oligomeric nature of meprin α confers a selection pressure on fetuin-B and supports the notion that meprin α oligomers exist in vivo.

## Inhibition of meprin α

Comparison between the cryo-EM reconstructions of the latent and activated states reveal subtle conformational changes about the M12A catalytic domain (Fig. 4a–c, Supplementary Table 1, Supplementary Fig. 11). The pro-peptide forms a small helical cork which plugs the active core and thus sterically occludes the active site (Fig. 4d, Supplementary Fig. 11). Cleavage of an exposed flexible loop, defined by residues L61-L67 located distal to the active site, releases the propeptide and exposes the catalytic core of the active site. Accordingly, the steric occlusion of the active site represents the major autoinhibitory mechanism. Upon activation, the two lobes of the M12A catalytic domain collapse around the active site (RMSD 3.3 Å), permitting the formation of a more tightly folded active site core (Fig. 4e).

The majority of known meprin inhibitors possess a hydroxamic acid moiety that targets the central zinc atom of the active site[64]. Thus, previous generations of inhibitors, e.g. actinonin or sulphonamide-based compounds, exhibit poor specificity and can lead to off-target effects by inhibiting other metalloproteases. In particular, there is a need for specific inhibitors of meprin proteases that target either meprin α or meprin β. Our group has previously developed a small molecule inhibitor of meprin α, compound 10d (3-[bis(1,3-benzo-dioxol-5-ylmethyl)amino]propane-hydroxamic acid)[44]. The compound also binds to the zinc via a hydroxamate moiety, but is built on a modified scaffold, i.e. a tertiary amine functionalised with two 1,3-benzodioxole groups. This modified scaffold leads to a high selectivity over other zinc-metalloproteases, i.e. MMPs and ADAMs. Compound 10d shows promising inhibitory activity and furthermore is specific to meprin α with an apparent potency (IC50) of 160 nM versus 2950 nM for meprin β.

To characterise the structural basis of binding and mechanism of inhibition by the small molecule inhibitor, we determined the structure of meprin α after co-incubation with compound 10d. Inspection of the active site revealed additional cryo-EM density positioned deeply within the groove of the active site (Fig. 4c, f). Chemical docking

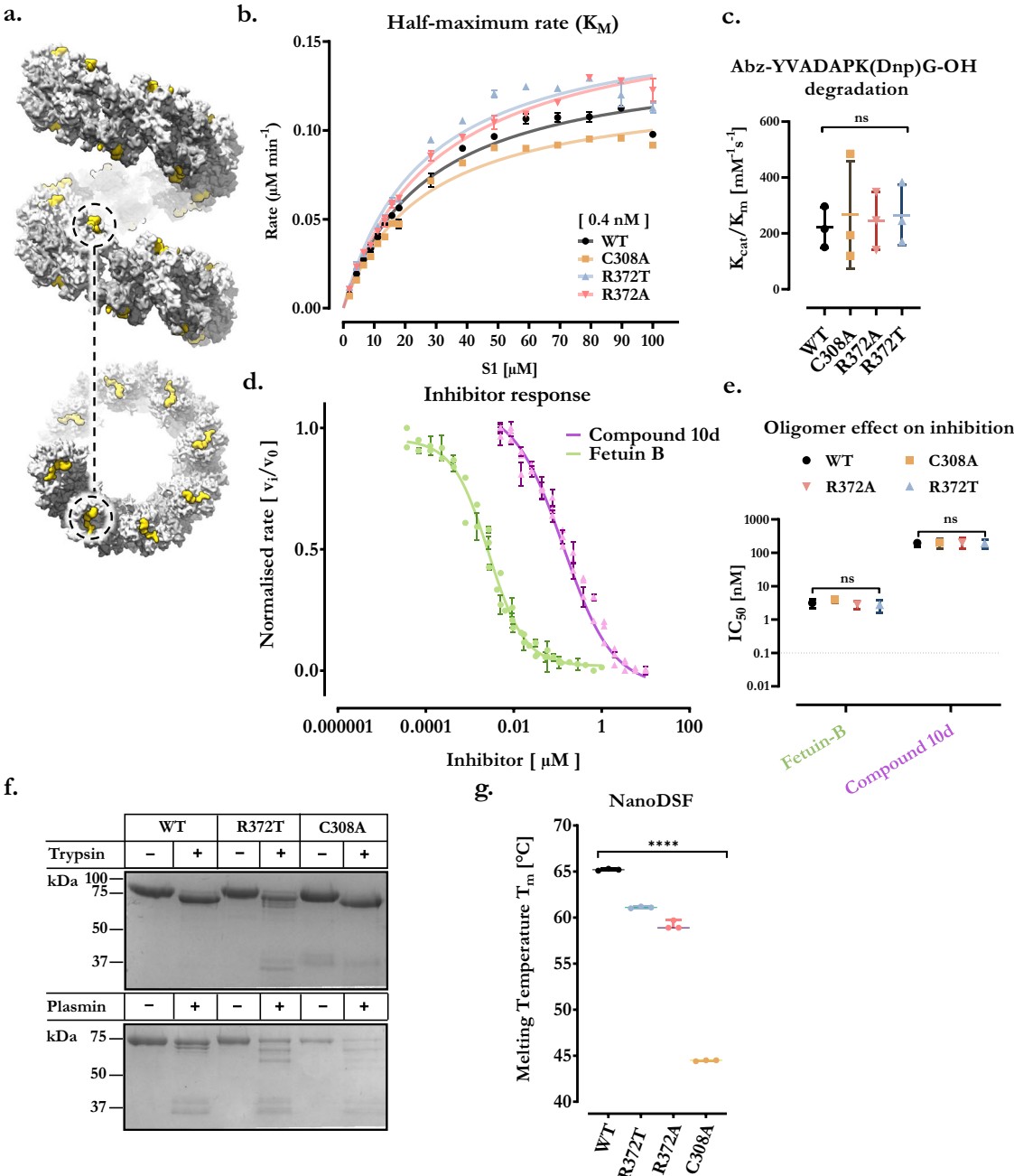

**Fig. 3 | Meprin α oligomers are proteolytically and thermally stable compared to lower-stoichiometry variants. a** Periodic arrangement of the active site and pro-domain are highlighted as yellow in the surface representation of an idealised meprin α helical segment. **b** Example curve representing a series of first-order rates (determined by fitting the linear region of fluorescence versus time) at a given meprin concentration (each point is the mean of $n = 2$ technical repeats; each curve was measured in $n = 3$ independent experiments). The $k_{cat}/K_m$ is determined from this graph for several meprin concentrations and in triplicate **c** First order rate constants ($k_{cat}/K_m$) of meprin α and variants for small fluorogenic peptide cleavage. Oligomeric state does not appear to affect rate of cleavage of small molecule substrate ($n = 3$ independent experiments with $n = 2$ technical repeats). Statistical test of one-way ANOVA, n.s. is not significant, Tukey's test for multiple comparisons. **d** Apparent potency measured as the inhibitory constant ($IC_{50}$) of meprin α inhibitors. Determined by fitting and normalising the linear region of fluorescence versus time, in the presence of varying amount of inhibitor at a fixed meprin α concentration (each point is the mean of $n = 2$ to 4 technical repeats across $n = 3$ independent experiments). **e** Globular proteinaceous inhibitor, murine fetuin-B, and small molecule compound 10d were unaffected by oligomeric state (determined by fitting 3d, i.e., $n = 3$ independent experiments). Statistical test of one-way ANOVA, n.s. is not significant, Tukey's test for multiple comparisons. **f** Proteolytic stability of meprin α and variants against trypsin and plasmin. Meprin α oligomers were more stable compared to lower-stoichiometric variants. **g** Meprin α thermal stability measured by nanoDSF ($n = 3$ independent experiments). Meprin α possess superior thermal stability compared to variants that lack either the disulphide bridge (most drastic) or helical interface interactions. One-way ANOVA reports statistically significant difference for each comparison (i.e., all versus all; $p < 0.0001$, ****). For all panels, each point is the mean and error bars represent the standard deviation (±σ). Source data are provided as a Source Data file.

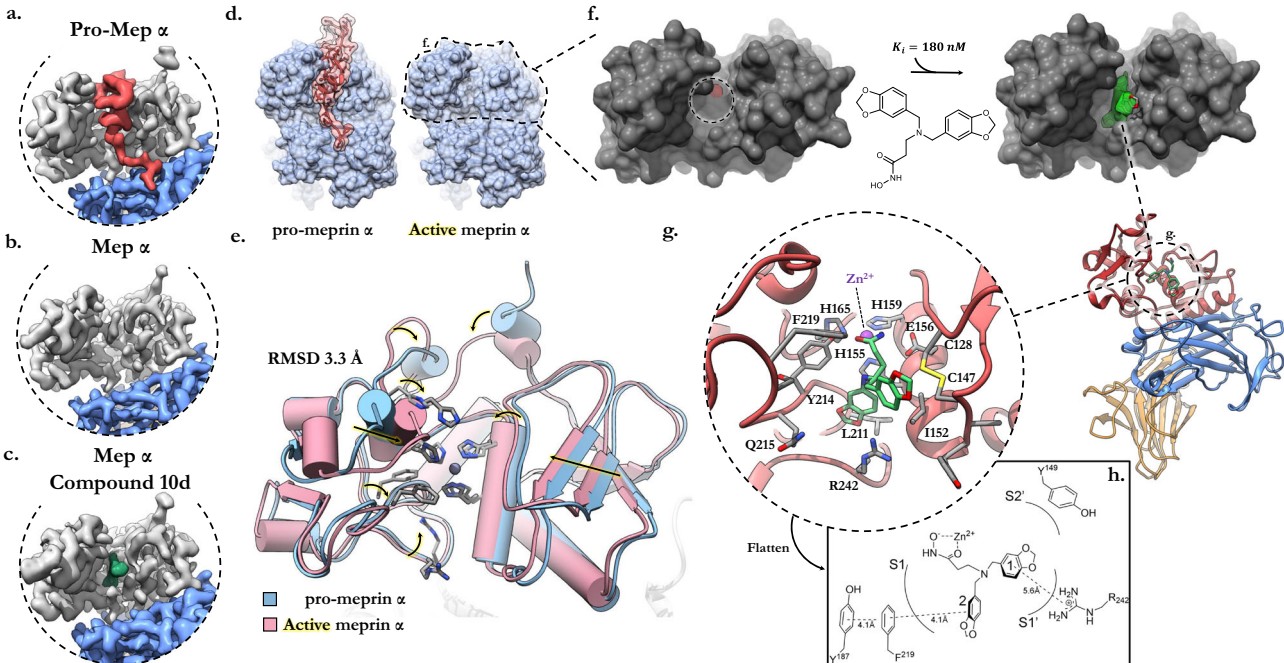

**Fig. 4 | Structural basis of meprin α auto-inhibition, small molecule inhibition and activation.** Zoomed view of the cryoEM reconstructions showing (**a**) the zymogen (pro-domain in red), **b** activated form (active site in yellow) and **c** drug inhibited form (compound 10d prototype drug in green). The M12A protease domain is coloured light grey, with the MATH domain in blue. **d** Surface representation of the pro and active form of meprin α focused on the active site cleft. **e** Meprin α activation requires cleavage of the pro-domain helical plug. Activation of meprin α is accompanied by conformational relaxation of the M12A protease domain resulting in a tightly folded active site core. **f** Surface representation of meprin α M12A protease domain highlighting (dashed circle) the active site and depth of active groove. The cryoEM density of compound 10d is shown in green, which is deeply buried within the cleft. **g** Focused view of the meprin α active site with the small molecule inhibitor docked in its inhibitory conformation. Interactions between the hydroxamate group and zinc core constitute the major steric mechanism for inhibition of the catalytic glutamic acid (E156). The 1,3-benzodioxole groups are positioned within the deep groove forming transient interactions with aliphatic, hydrophobic and backbone functional groups. **h** Projected two-dimensional plot of the active site and compound 10d interactions.

simulations were used to flexibly fit the ligand structure into the density, revealing interactions between the hydroxamic acid group and putative zinc cation core of the catalytic triad as expected (Fig. 4f, g). The two 1,3-benzodioxole groups provide further supporting interactions, predominantly from hydrophobic interactions. One benzodioxole group is deeply buried within the S1' pocket forming an anchor point, while the other sits flush against the central helix of the M12A active site. Notably, due to the large width of the M12A groove, which accommodates the entire pro-domain α-helix, relatively few contacts are present between the M12A lobes and compound 10d. As such, we observe some degree of inhibitor mobility as assessed by loss of ordered cryo-EM density. Nevertheless, in this position both the catalytic zinc and binding pocket are sterically occluded by compound 10d, thereby inhibiting the proteolytic function of meprin α (Fig. 4g).

Despite the selectivity of compound 10d for meprin α most contacts of the interaction are strongly conserved between meprin α and meprin β (Supplementary Fig. 12a, b). The only exceptions being I152 (T149 in meprin β) and Q215 (S212 in meprin β). As a result, it remains unclear at the current resolution how compound 10d derives its selectivity for meprin α. Notably, however, R238 of meprin β (R242 in meprin α) adopts a distinct rotamer position relative to meprin α, due to charge repulsion with R146 (Y149 in meprin α). In this position, R238 of meprin β sterically occludes the binding of compound 10d (Supplementary Fig. 12a, c). Further modification of compound 10d to exploit this charged residue and to increase contact surface area with the M12A catalytic groove may be beneficial in terms of improving affinity and specificity to meprin α.

## Discussion

Proteases are crucial players in fundamental cellular processes in human health such as proliferation, differentiation, inflammation, and ECM homeostasis, which drive various pathophysiological states when dysregulated. For example, degradation of the ECM is a typical hallmark of aggressive metastatic cancers by enabling cell migration. Meprin proteases function in the epithelium to regulate ECM homeostasis while having both anti- and pro-inflammatory roles in cell signalling[25]. Indeed, meprin α has been implicated as an oncogenic factor which, when highly expressed, is correlated with more metastatic and aggressive forms of certain colorectal cancers. Further elevated levels of meprin α are observed in connection with nephritis[65–67].

Current available drugs to meprin α bind with high affinity but lack high specificity. They also inhibit the functions of meprin β. Therefore, efforts to engineer more selective and potent meprin α inhibitors are ongoing. While the structure of meprin β has been described previously, the structure of meprin α was unavailable. Structures of meprin β have provided mechanistic insights and supported rational design efforts to develop specific inhibitors to this homologue. Conversely, since the structural differences of these proteases underpin the selectivity of compounds, the absence of a meprin α structure has obfuscated drug discovery programs.

Previous studies report meprin α forms large, heterogeneous oligomeric species. We therefore used an electron microscopy approach to elucidate the structure. The high-resolution structures of the meprin α oligomer were determined in four key states. The zymogen form of meprin α is plugged by an α-helical pro-peptide that sterically occludes the active site. Upon proteolytic cleavage of the pro-peptide the active is exposed and the M12A protease domain undergoes a conformational relaxation. The exposed active site defines a deep pocket that positions the catalytic glutamic acid in close proximity to the substrate peptide as described for meprin β[56]. Structural studies of meprin α in complex with a prototype selective inhibitor revealed the mode of inhibition. Furthermore, these data

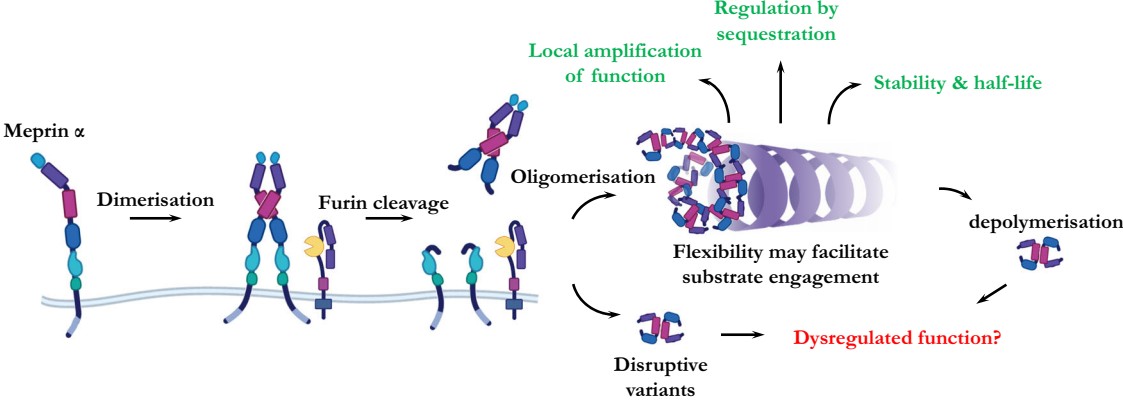

**Fig. 5 | Proposed mechanism of action of meprin α helices.** Monomeric full length meprin α dimerises during expression and is proteolytically shed from the membrane bilayer via furin-protease activity. Shed meprin α homodimers oligomerise via interactions of MAM/MATH domains forming large oligomeric helices. These helices function to locally amplify proteolytic activity as well as regulate the extent of activity by sequestrating the protease to specific cellular and tissue locations. Further, oligomerisation putatively improves in vivo stability and half-life. Disruptive mutations that interfere with helix formation or increase rate of depolymerisation are postulated to drive dysregulated and detrimental meprin α function.

reveal subtle differences between meprin α and meprin β that may underpin inhibitor specificity. As expected, the prototype inhibitor was observed to competitively bind to the active site thereby blocking substrate engagement. These data provide long sought-after details of the meprin α active site for drug discovery programs and offer insight into the unusually large oligomeric form of meprin α.

Intriguingly, meprin α dimers associate to form giant left-handed helical filaments of seemingly no restriction to length in vitro. Protein oligomerisation drives many biological processes such as allosteric control of activity, regulation of protein activity by spatial sequestration, control of local concentrations, increased stability against denaturation[68,69], and more. As such, formation of oligomers offers functional, genetic, and physicochemical advantages over monomeric counterparts. A prominent example of this includes oligomeric tripeptidyl peptidase II (TPPII), a cytosolic dimeric enzyme classified as the largest peptidase complex so far[70]. But, while TPPII forms a spindle-like rigid structure, the meprin α helix is flexible with no further contacts between neighbouring turns. In TPPII this rigid structure results in restricted access to the active site and the peptidase acts only on smaller und unfolded substrates and was shown to have 10-fold increased activity after oligomerization. Overall, in vitro no such correlation could be found for oligomeric meprin α compared with lower-stoichiometric variants indicating that the functional role of the oligomerisation, if any, may only be apparent in vivo. In contrast, stability assays indicate that meprin α oligomers are less susceptible to proteolytic degradation and have improved thermal stability when compared to lower stoichiometry forms. The increased stability against proteolytic degradation seems to be meaningful considering the secretion of meprin α into compartments with high proteolytic potential such as the lumen of intestine and proximal kidney tubuli.

Lastly, it has been established that differential recognition and cleavage of substrates is dependent on enzyme localisation, such as membrane bound versus soluble meprin β. As is the case with TPPII, the formation of a helix could also function to sequester and concentrate activity to a localised region[71,72]. The effect of localised activity would not be captured in our in vitro assays, however in vivo it may be important for the catabolic function in the lumen of the intestine and proximal kidney tubuli, and additionally, for the controlled degradation of the extracellular matrix. Intriguingly, in some tissue contexts such as the epidermis, meprin α expression is observed to be specifically localised in particular subregions (e.g. the stratum basale)[14]. Owing to their size, meprin α filaments may have limited capacity to diffuse into neighbouring tissue areas thereby restricting the ECM remodelling activity to appropriate regions.

Considering these finding and our structures, we mapped frequently mutated residues that are associated with disease to the domains that control meprin α oligomerisation. This localisation suggests that these mutations may drive disease by destabilising the oligomeric form. For example, by the release of higher proportions of freely diffusing meprin α dimers that exert function in a dysregulated manner (Fig. 5). The restricted helical arrangement further suggests that some activators, substrates, and inhibitors may have limited accessibility to the meprin α pro-peptide and active site. However, we were unable to detect any significant differences between meprin α oligomeric forms when assessing small molecule and globular substrates, nor small molecule and globular inhibitors. Indeed, structural studies of human fetuin-B in complex with meprin α are consistent with crystal structures of murine fetuin-B and meprin β[73]. Unlike meprin β, these inhibitory complexes were observed to form supramolecular helical packing as a result of the intercalated arrangement of meprin α and numerous fetuin-B molecules.

Taken together, our findings suggest that differences between stoichiometric forms of meprin α may depend entirely on their localisation which, consequently, dictates their distinct activity profiles[14,52]. An alternative hypothesis is that the oligomeric ultrastructure somehow mediates interactions with certain substrates or has a scaffolding role. Ultimately the impact of these effects on biological function, if any, remains to be shown. It is currently unclear what role meprin α oligomerisation mediates at a cellular level. Previous studies suggest the tissue and cellular localisation of meprins underpins both correct function and regulates activities of these key proteases[14,52]. Therefore, we suggest meprin α oligomerisation may be a regulatory mechanism that functions to either stabilise, sequester and/or locally amplify proteolytic activity (Fig. 5). Further studies in these regards will be informative in understanding the role of meprin α in cancers and other human disease, for example, studies in vivo to assess the physiological effects of meprin α depolymerisation and the importance of meprin α oligomerisation in normal ECM homeostasis.

## Methods
### Expression, purification, activation
Expression of pro-meprin α (and variants) was achieved in Schneider-2 *Drosophila* cells (S2 cells). Briefly, the sequence of pro-meprin α (V22-S600, i.e. omitting the native signal peptide; uniprot Q16819) containing an N-terminal Strep-Tag (WSHPQFEK) was cloned into pMT/BiP/V5 with a C-terminus stop codon after residue S600, enabling stable cell lines to be produced (Supplementary Table 2). The pMT/BiP/V5 vector encodes the BiP signal peptide enabling meprin secretion

with an N-terminal Strep-Tag for purification. The pro-meprin α variants C308A, R372T and R372A were produced by site-directed mutagenesis (Supplementary Table 2). To induce the production of pro-meprin α, S2 cells were grown in Schneider's Drosophila Medium (Biowest) supplemented with 1 mM copper sulphate and 0.05% Pluronic™ F-68 at 28 °C and 80 rpm for two days. The supernatant was harvested by centrifugation and immediately purified by hydrophobic interaction chromatography applying expanded bed adsorption (HIC-EBA, equilibration buffer 30 mM Tris HCl pH 7.4, 1.5 M ammonium sulphate; elution buffer 30 mM Tris HCl pH 7.4, 100 mM NaCl). The eluate of HIC-EBA was subjected to affinity chromatography using Strep-Tactin® column (5 ml cartridge, GE Healthcare Life Science, equilibration buffer 30 mM Tris HCl pH 7.4, 100 mM NaCl; elution buffer 30 mM Tris HCl pH 7.4, 100 mM NaCl, 2.5 mM desthiobiotin). Finally, pro-meprin α was activated by trypsin cleavage applying immobilized trypsin on magnetic beads (PT3957-1, Takara, buffer 30 mM Tris HCl pH 7.4, 100 mM NaCl, 5 mM $CaCl_2$). Protein purity was assessed by Coomassie Blue SDS-PAGE, dot immunoblot or Western blot (probed with either polyclonal goat (#AF3220) or monoclonal mouse (#MAB3220) anti-meprin α at 1 μg mL$^{-1}$ final concentration).

### Animal work
Mouse urine (from standard BL6 mice) was collected in metabolic cages for 24 h. The animal experiment was approved by the responsible animal ethics committee of the state of Saxony-Anhalt, Germany (Landesverwaltungsamt Sachsen-Anhalt, Department of Consumer Protection and Veterinary Affairs, Halle (Saale), Saxony-Anhalt, Germany) under the following approval number: 42502-2-1473 MLU.

### Kinetic analysis
The determination of enzymatic activity was based on the cleavage of the fluorescent peptide substrate Abz-YVADAPK(Dnp)G-OH. Measurement of kinetic parameters was performed in 384 well black plates in a volume of 60 μl (assay buffer 50 mM HEPES, 150 mM NaCl, pH 7.4, 0.05% Brij). Enzyme solution in buffer was applied and subsequently substituted with inhibitor/DMSO normalisation (Digital Dispenser D330e, Tecan, Switzerland) and preincubated at 30 °C for 20 min. After addition of substrate reaction was measured at excitation/emission wavelength 340/420 nm on a plate reader (Clariostar, BMG Labtech, Germany). For estimation of first order rate constants of $k_{cat}/K_m$ substrate concentrations from 0.8 to 83 μM were used. Concentration of active enzyme was determined to be 3–5 nM by titration of the active site using a tight binding inhibitor. The apparent potencies ($IC_{50}$) were estimated at a substrate concentration of 10 μM varying inhibitor from 10 μM to 5 nM (compound 10d) and 200 nM to 0.03 nM (fetuin B). Kinetic parameters were determined at least in triplicates in separate experiments and evaluated with GraphPad Prism software (Graphpad Software, Inc., USA). Mouse fetuin B was a generous gift from Hagen Körschgen and Walter Stöcker (University Mainz).

### Turnover of protein substrates
Different enzyme-substrate ratios of meprin α wild type and meprin α C308A were incubated with human tropoelastin, human fibronectin and rat collagen I. In the case of tropoelastin (kindly gifted by Mathias Mende, group of Prof. Pietzsch, Martin-Luther University Halle-Wittenberg), the reaction was carried out in 50 mM Tris-HCl pH 7.4 buffer at 37 °C and molar ratios of enzyme to substrate of 1:10$^4$ and 1:10$^5$. Samples (10 μg) were removed every 30 min for of 3 h. The cleavage of fibronectin (ab209886, Abcam) was assessed in 50 mM HEPES pH 7.4 buffer containing 150 mM NaCl. Enzyme to substrate ratios of 1:10$^4$ and 1:10$^5$ were tested. Reaction conditions were as described above. Collagen I from rat tail was prepared according to Gorisse et al[74] and the reaction was carried out in 50 mM HEPES pH 7.4 at 37 °C at a ratio of 1:10$^4$. All cleavage products were evaluated by reducing SDS-PAGE, followed by Coomassie-staining.

### Proteolytic stability
A total of 9 μl of enzyme (2 μM) were supplemented with 1 μl Trypsin/Plasmin (2 μM) or buffer and incubated at room temperature. After 20 min, reactions were stopped by addition of 2.5 μl 5-fold reducing sample buffer. Samples were heated and subjected to analysis by 10% SDS-PAGE.

### NanoDSF
NanoDSF measurements were conducted on a Prometheus NT.48 (NanoTemper) instrument. Each meprin α variant was standardised to a protein concentration of 0.1–0.3 mg ml$^{-1}$ in 30 mM Tris HCl, 100 mM NaCl, pH 7.4. Thermal unfolding was monitored by the intrinsic tryptophan fluorescence at emission wavelengths 350/330 nm in 1 °C min$^{-1}$ increments from 20 °C to 95 °C. The apparent melting temperature ($T_m$) was determined as the maximum of the first derivative of the 350/330 nm ratio. Each variant was measured in three independent experiments.

### MADLS / SEC-MALS
Multiangle dynamic light scattering analyses of meprin α was performed using a Zetasizer Ultra (Malvern Panalytical Ltd., UK) and He-Ne laser at 632.8 nm and constant power of 10 mW at 30 °C. Concentration of protein was 0.3 mg ml$^{-1}$ in 30 mM Tris, 100 mM NaCl, pH 7.4. Detector angle for polydispersity index (PI) was 173°. The results are presented as the average value of three to five experiments. For size exclusion-multiangle dynamic light scattering, meprin α C308A (100 μl at 0.74 mg ml$^{-1}$) and meprin α R372T (100 μl at 0.44 mg ml$^{-1}$) were applied onto an AdvanceBio SEC column (300 Å, 7.8 × 300 mm, Agilent Technologies). The data was collected using flexControl (Compass software suite) and evaluated using ASTRA® 6 (Wyatt Technology).

### MALDI-TOF-MS
For MALDI-TOF mass spectrometry all samples were purified using C4 ZipTip® pipette tips (Merck) before analysis as described in the supplied manual. The purified sample (1 μl) was mixed with 1 μl of DHAP-matrix (7 mg 2,5-dihydroxyacetophenone, 375 μl ethanol, 125 μl of 16 mg ml$^{-1}$ di-ammonium hydrogen citrate) and 1 μl of 0.1% TFA and applied onto a metal target plate. The samples were analyzed in linear positive mode (LP_30-210 kDa according to Bruker Daltonics) using an AutoflexTM speed MALDI-TOF/TOF device (Bruker Daltonics). Protein Calibration Standard I and II (Bruker Daltonics) were applied for calibration of the device.

### Mass photometry
Mass photometry was conducted on an active anti-vibration platform using a TwoMP instrument (Refeyn; 20 °C). Meprin variants were measured at a final concentration of roughly 2–5 nM and calibration standards were measured on the day of the experiment. Mass photometry images were acquired and analysed using the Refeyn Aquire$^{MP}$ and Discover$^{MP}$ packages (v2.5) respectively.

### Cryo-EM and cryo-ET sample preparation and data collection
For tomography samples, Quantifoil Cu R 2/2 grids were glow-discharged for 30 s in a Pelco EasyGlow, and 3 μL of meprin α (0.5 mg ml$^{-1}$ in 30 mM Tris buffer, 100 mM NaCl, pH 7.4) was mixed with 10 nm gold nanoparticles and was applied to the glow discharged surface and blotted at 4 °C and 100% relative humidity for 3 s and a blot force of −3 using the Vitrobot IV System (Thermo Fisher Scientific). Grids were plunged in 100% liquid ethane. The grids were stored under liquid nitrogen until TEM data collection. Tilt series were acquired at 300 kV on a FEI Titan Krios G1 and digitised on a postGIF K2 Summit Direct Electron Detector. A dose-symmetric tilt acquisition scheme was used with 3° increments, an electron dose of 2.6 e$^-$ A$^{-2}$ per tilt. Micrographs were acquired in dose fractionation mode with 0.5 e$^-$ A$^{-2}$ frame$^{-1}$ at a pixel size of 0.186 nm × 0.186 nm.

Similarly for single-particle cryo-EM, initial grid freezing conditions were tested and screened on a Tecnai T12 electron microscope (Thermo Fisher Scientific). Blotting was carried out as above, with the following modifications, a blot time of 2.5 s, blot force of −5 and drain time of 1 s were used. The grids were stored under liquid nitrogen until TEM data collection. Briefly, dose fractionated movies were collected on a Titan Krios (Thermo Fisher Scientific), equipped with a Quantum energy filter (Gatan) and Summit K2 (Gatan) or K3 (Gatan) direct electron detector. Data acquisition was performed using either SerialEM[75] or EPU (Thermo Fisher Scientific). Meprin α/fetuin-B complex was made immediately prior to freezing, by mixing stoichiometric quantities (final concentration 1 mg ml⁻¹) and allowing to incubate for 3 minutes at room temperature. His-tagged recombinant human fetuin-B (11834-H08H) was purchased from SinoBiological. Similarly, compound 10d was added in excess (-8 fold $K_i$) and allowed to incubate for 1 minute, prior to freezing.

### Cryo-EM and cryo-ET image analysis (SPA and STA)

Dose fractionated movies were firstly compressed to LZW TIFF format with IMOD[76] to save disk space. Correction of beam induced motion and radiation damage was performed with MotionCor2 (v1.3.0)[77]. Corrected frames were dose-weighted and averaged for all further processing. Tilt series were aligned and backprojected using IMOD[76,78,79], and CTF was estimated in Warp[80]. Subtomogram averaging was performed using Dynamo[81] or Relion-4[82,83]. Particle picking was conducted manually in Cube[80]. Sub-tomograms averages were generated according to the Relion-4 pipeline. An ab initio volume was generated and subsequently refined to 2.6 nm. Subsequent 3D classification and additional refinements, including tilt series and CTF refinements, yielded a 1.7 nm resolution map. Finally, signal subtraction and local refinement of a smaller segment resulted in an improved reconstruction with nominal resolution of 1.27 nm. DeepEMhancer was used to sharpen this final STA. A single sub-tomogram volume (extracted from a tomogram in chimera by 'vop box') was later used as initial volume for single particle analysis (SPA).

Despite exhaustive efforts, we were unable to obtain meaningful high-resolution reconstructions using traditional workflows (for SPA and STA), we suspect due to flexibility. We ultimately solved the problem with a pseudo-SPA approach, where segments (and sub-regions) of the filaments were treated as if they were isolated single particles. Approximately 20 nm segments of the filament were reconstructed using helical processing (without symmetry) generating a 10–12 Å resolution reconstruction, which suffered from errors in alignment due to global flexibility (Supplementary Fig. 2a). This reconstruction enabled non-overlapping sub-regions consisting of two meprin α dimers to be localised within the original extracted particles with roughly nanometre precision (Supplementary Fig. 2b, c). High resolution refinement of these sub-regions (to 2.4–3.4 Å resolution) was achieved by treating these as independent single particles after masking and subtracting the signal of neighbouring segments. In doing so, we thereby overcame the prohibitive alignment errors resulting from continuous conformational heterogeneity of the filament (Supplementary Fig. 2d, e). In summary, segments of particles were initially analysed using helical processing to generate a consensus reconstruction, this was then used to guide the extraction of localised sub-particles which were subsequently treated as independent particles for SPA. Aspects of this approach are similar to those employed to study bacterial flagella[84,85].

Estimates of CTF were performed in CTFFIND (4.1.13)[86] or Warp (1.0.7)[80]. A combination of manual and automated filament particle picking was performed by hand or with crYOLO (1.8.1)[87,88] operating in filament mode. A small, hand-picked subset of images were used to train crYOLO. Particles were extracted within 400-pixel boxes and normalised within RELION (v2.1-v3.1)[89–91]. Initial rounds of 2D classification in cryoSPARC[92,93] and RELION were used to discard malformed

particles and poor-quality images. The sub-tomogram average was used as an initial volume to bootstrap the SPA refinement. Asymmetric helical refinements were performed in RELION. Flexibility of meprin α resulted in large-scale, continuous conformational heterogeneity between the images. This was apparent in 2D class averages which displayed clear secondary structure features at the centre of the averages, with diffuse signal and poor coherency that worsened moving away from the centre along the helix lengthwise. Three-dimensional variability analysis in cryoSPARC revealed numerous modes of flexibility[94]. This also prevented symmetry determination by layer line analysis. Resultantly, all the refinements of the full helical assembly that were attempted, gave rise to severely limited reconstructions with resolution on the order of nanometres.

The best reconstructions of the full filament from C1 helical refinement were used to guide the placement of meprin β dimer modes (PDB: 4GWN)[56]. Regions of the helix that corresponded to dimer pairs were assigned vectors from the centre to the dimer pairs in UCSF Chimera (v1.13), giving rise to between 6 and 8 points of interest (according to Ilca et al.[95]). A mask of approximately two dimers was used to isolate regions of interest from the reconstruction. A small space was left between regions as a buffer to accommodate some variation in the placement of models (and prevent particle duplication due to overlap). These regions of interest were individually removed from the reconstruction, to generate masks of the remainder of the helix. These masks were subsequently used for partial signal subtraction and localised particle extraction within a modified version of *localised_reconstruction.py*[95] or directly within RELION.

Localised reconstruction of these portions of helix gave rise to a notably more homogeneous volume that displayed secondary structures. Subsequent 2D and 3D classification of these segments in RELION or heterogeneous refinement in cryoSPARC facilitated the removal of poor-quality images. Homogeneous refinements in cryoSPARC were performed to obtain optimised alignment and offset parameters. Peripheral regions of the localised reconstruction corresponding to the boundaries of the subtracted regions were noisy and diffuse. Therefore, these areas were subtracted and further rounds of local refinement in cryoSPARC were performed. CTF parameters were refined and corrected in cryoSPARC or RELION including magnification anisotropy, beam tilt, trefoil, tetrafoil and astigmatism[90]. Finally, non-uniform refinement was employed to account for variation in resolution across the reconstruction that gave rise to minor errors in alignments[93].

For reconstructions of fetuin-B/meprin α, RELION was unable to solve the global alignment problem. Therefore, asymmetric reconstructions of the whole helix were performed in cryoSPARC. Localised sub-particle extraction was performed in RELION and global alignment of the extracted subregions was performed in RELION (here cryoSPARC failed to solve the global alignment problem). Finally, local non-uniform refinement in cryoSPARC with gaussian prior yielded the final reconstruction. These final particles and optimised metadata were exported from cryoSPARC for 3D classification within RELION. Low resolution signal dominated and therefore the CTF was ignored until the first zero giving rise to markedly improved classification.

For all maps, local resolution was estimated in RELION with a windowed FSC. Map sharpening was performed in cryoSPARC, RELION[96] or with deepEMhancer[97]. Various amplitude corrected maps were employed for model building. Conversion between cryoSPARC and RELION were performed with *pyem*[98].

### Model building

An initial model of full length pro-meprin α were generated via homology with meprin β (PDB-4GWN)[56] using the SWISS-MODEL server[99] and rigid body fit into the cryo-EM reconstruction with UCSF Chimera (v1.13)[100]. Subsequently, the model was flexibly fit and interactively refined in ISOLDE[101], ChimeraX (v1.3)[102] and Coot[103,104]. Where possible, N-linked glycosylations were modelled in Coot. Models of the

active form and inhibitor-bound form were generated similarly, using the model of pro-meprin α as a starting template. Lastly, molecular docking analysis was carried out using the fetuin-B/meprin β crystal structure superimposed onto the meprin α helix[73]. Subsequently, a model of the meprin α/fetuin-B complex was built into the cryoEM density. Initially, we took the model of active meprin α and the AlphaFold model of human fetuin-B[105,106]. These were rigid body fit into the cryo-EM reconstruction and were subsequently flexibly fit and interactively refined in ISOLDE/ChimeraX using secondary structure and all atom constraints to prevent the model from diverging. All models were subject to real space refinement and validation in PHENIX[107,108] and MolProbity[109].

### Reporting summary
Further information on research design is available in the Nature Research Reporting Summary linked to this article.

## Data availability
The Cryo-EM maps of the structural data generated in this study have been deposited in the Electron Microscopy Data Bank (EMDB), while atomic coordinates are available from the RCSB Protein Data Bank under the following accession codes. Pro-meprin α (tetramer), EMD-26419 and PDB-7UAB. Pro-meprin α (single subunit), EMD-26420 and PDB-7UAC. Full meprin α helix in the active state (C1 reconstruction), EMD-26421. Meprin α in the active state (single subunit), EMD-26422 and PDB-7UAE. Meprin α in complex with small molecule inhibitor (tetramer), EMD-26423 and PDB-7UAF. Meprin α in complex with the native fetuin-B inhibitor (tetramer), EMD-26426 and PDB-7UAI. Full meprin α helix in complex with the native fetuin-B inhibitor (C1 reconstruction) EMD-26424. Sub-tomogram average of pro-meprin α (12-subunits), EMD-27689. The following atomic coordinates were used for analysis, PDB-4GWN, AF-Q16819-v2, and AF-Q9UGM5-v2. Source data for Fig. 3 b-g, Supplementary figures 1 c-e, and 9, are provided with this paper. Mutagenesis data was obtained from the COSMIC database, meprin α [https://cancer.sanger.ac.uk/cosmic/gene/analysis?ln=MEP1A] and meprin β [https://cancer.sanger.ac.uk/cosmic/gene/analysis?ln=MEP1B]. Sequence numbering refers to the full length meprin α available from UniProt Q16819. Any additional information and data are made available by request from the corresponding author. Source data are provided with this paper.

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

## Acknowledgements

C.B.J. acknowledges the support of the Australian Government RTP scholarship. We thank the staff of the Monash Ramaciotti Centre for Electron Microscopy, the support of the MASSIVE supercomputer team and the Monash Macromolecular Crystallisation Facility. This project was supported by BMBF (APRA program), Germany (D.S., S.S.). We gratefully acknowledge the organizational support by Vivoryon N.V.; the valuable scientific discourse and proofreading by Stefan Tasler; and the mass spectrometry support and contributions of Marcel Neumann (Fraunhofer IZI-MWT).

## Author contributions

D.S., C.F., M.W. expressed, purified, and analysed protein. D.S., C.F., C.J.L. prepared and screened cryo-electron microscopy samples. C.B.J.,

C.J.L., D.S. and C.F. processed and analysed cryo-electron microscopy data. C.B.J., C.J.L., D.S. and C.F. built and analysed protein models. C.B.J. wrote the first draft. C.B.J., D.S., C.F., C.J.L., J.C.W. co-wrote, revised, and edited the manuscript. C.B.J., D.S., C.J.L., produced figures. H.V., A.dM., C.J.L. collected cryo-electron microscopy data. D.R. and C.J. performed inhibitor docking and structural analysis. C.B.J., D.S. and C.F. produced figures. D.S., S.S., and J.C.W. acquired funding and provided supervision. All authors contributed to the discussion and interpretation of results.

## Competing interests

C.J. is an employee of Vivoryon Therapeutics N.V., Halle (Saale), Germany. The remaining authors declare no competing interests.
