## [Peer Review File · Nature Communications]

Helical ultrastructure of the metalloprotease meprin α in complex with a small molecule inhibitorReviewer #1 (Remarks to the Author):

Manuscript#: NCOMMS-22-09816

“Helical ultrastructure of the oncogenic metalloprotease meprin α in complex with a small molecule hydroxamate inhibitor”.

Summary: The authors utilized cryo-electron microscopy to determine the high-resolution structure of the zymogen and mature form of meprin α , as well as the structure of the active form in complex with a prototype small molecule inhibitor and human fetuin-B. The authors also used mutagenesis to examine a role of meprin α oligomerization in its function and regulation and found that oligomerisation improves proteolytic and thermal stability but does not impact substrate specificity or enzymatic activity.

Comments: This is a very interesting, original, and important study for the meprin field. To the best of my knowledge, this is the first high-resolution structure of meprin α alone, and which is especially important, in the complex with the selective meprin α inhibitor 10d. The complex with meprin α inhibitor will enable further inhibitor/drug discovery studies, hopefully resulting in novel drugs.

The claims of proteolytic stability of meprin α oligomers and monomers are not sufficiently substantiated (see issues below). The rest of conclusions are well supported. No flaws in data analysis were observed. The methodology is sound, and the overall work is of high quality.

Issues:

Major:

- 1. Figure 3f. Trypsin treatment appears to produce smaller molecular size meprin α in both oligomeric and monomeric variants. C308A mutant which runs at 70kDa in both native and reducing conditions (Suppl fig 1C,D) runs at ~100kDa in Fig 3f. What is the reason for this inconsistency?**
- 2. This reviewer agrees that at the time frame of proteolytic stability assay (20min) WT enzyme appears to be more stable. However, why the proteolytic digest was stopped after 20 min? Perhaps, if it was run longer there would be no or little difference between WT and mutants. Both C308A and R372T mutants appear to have digest products at ~50kDa after trypsin treatment and WT, C308A and R372T have ~50kDa products after plasmin treatment, which suggests that neither meprin α oligomers nor monomer are 100% proteolytically stable.**

Minor:

- 1. The conclusions section would be helpful.**
- 2. Which tight-binding inhibitor of meprin α was utilized for determination of active enzyme? To the best of our knowledge, there are no tight-binding inhibitors of meprin α in the literature.**
- 3. The authors mention that off-target inhibition of proteolytic activity is a current limitation of existing candidates. However, Wang 2021 et al., paper (PMID: 33673639) lists meprin α inhibitor with more than 100-fold selectivity vs meprin β and related zinc-metalloproteases.**
- 3. Figure 3d. IC50 values are not constants as they are heavily dependent on the assay conditions, but rather estimates of true potency. The preferred terminology for IC50 values is “an apparent potency”.**
- 4. It is not entirely clear whether the meprin α structure was solved in complex with 10d and fetuin or ligands were docked into the meprin α structure. Different claims are made throughout the paper, therefore, it needs to be better explained.**

Reviewer #2 (Remarks to the Author):

In this manuscript, the authors report on the high-resolution quaternary structure of the metalloprotease meprin α , resolved by cryo-electron microscopy.

Meprin α has been reported in numerous organs and cell types like the intestine, the kidney, the skin and different leukocyte populations. Dysregulated expression, localization and activity of meprin α has been shown to be implicated in pathophysiological processes of various inflammatory diseases like inflammatory bowel diseases (IBD), acute kidney injury, Kawasaki syndrome (a systemic vasculitis of unknown etiology in childhood) and sepsis. The substrate pool of meprin α comprises numerous inflammatory mediators including cytokines, chemokines, extracellular matrix molecules, immune-regulatory adhesion molecules and receptors. Therefore, meprin α has been investigated as putative pharmacological target, and this requires structural information for proper drug design. Previous studies analyzing recombinant and purified meprin α indicated already huge meprin α homooligomers, determined by analytical gel filtration and transmission electron microscopy, up to 6 MDa in size. However, the real molecular assembly and identification of interaction sites for oligomerization were ambiguous. Unlike the close relative meprin β , which is exclusively forming dimers that allowed for resolving the crystal structure of the protease, identification of the oligomeric structure of meprin α requires different strategies. Herein, the authors revealed that meprin α builds a flexible, left-handed helical assembly, with a rather stringent diameter of about 22 nm but various length up to 500 nm, which represents a 27 MDa complex. This makes meprin α by far the largest secreted proteolytic enzyme. Additionally, the authors generated several variants of meprin α that nicely confirmed the amino acid residues responsible for oligomerisation (particularly Arg372) and furthermore provided structural data of meprin α in complex with the naturally occurring inhibitor fetuin B and a small synthetic molecular compound.

This paper merits publication in the journal Nature Communications, because it reports on the structural properties of the largest extracellular protease meprin α , building a fascinating helical structure that provides locally focused massive proteolytic activity, which certainly has specific biological function. This manuscript is the basis for many follow-up stories that aim to unravel these functions.

However, there are several points that need to be addressed prior publication.

Major issues:

- 1. For the experiments in Fig. 3f and Suppl. Figures 1e, 6a-c, 7 the meprin α variant R372A should be included. This is the perfect control, as it solely builds dimers.**
- 2. For Suppl. Fig. 1e standard proteins should additionally be loaded to better evaluate the different oligomeric states of meprin α .**
- 3. Lines 228-229: The authors stated that the meprin α variant C308A is not capable of forming dimers anymore, due to the loss of the stabilizing disulphide bridge. Here, the authors should explain in more detail, how the oligomers seen in Suppl. Fig. 6d can assemble.**

In this regard, it obviously happened that the spectra for meprin α C308A displayed in Suppl. Fig. 6a were cropped. Please show the entire graphs.

- 4. Since only one amino acid (Arg372) is crucial for the ionic interaction of the head-to-tail oligomer, the authors should provide data in how far salt concentrations and pH differences influence stability and size of the helical structure. For instance, the detected size of meprin α in urine appears to be smaller than the size of rec. enzyme (Suppl. Fig. 1e).**

5. Can the authors predict a model, if helical oligomerisation of meprin α could theoretically also start from a meprin α /meprin β heterodimer at the cell surface? That would consequently result in massive local meprin α activity at the plasma membrane.

6. The authors stated that the identified helical interface in meprin α is missing in meprin β . The alignment in Fig. 2c should be extended for additional species. It would be interesting to see, whether the capability of meprin α to form huge oligomers is preserved throughout evolution of the enzyme, which would underline the likely biological impact.

7. Fig. 4 and Suppl. Fig. 10: The part concerning the small molecule inhibitor is not fully convincing to me. Is the orientation of the small compound really fully resolved in the

cryo-EM structure? The explanation for the specificity towards meprin α over meprin β should be further defined. From the picture shown in Suppl. Fig. 10 the Arg238 in meprin β may adopt several other rotamer positions that would not interfere with the inhibitor.

Additionally, ovastacin, another astacin metalloprotease, is also inhibited by fetuin B. Does compound 10d inhibit this protease too, and if not, what are the molecular clashes that prevent the binding.

Minor issues:

1. The title could be edited with a stronger focus on the unique structural properties of meprin α . The 'oncogenic' function of meprin α is not experimentally addressed in this manuscript and other biological activities, e.g. in the brush borders of the intestine and kidney, might be more relevant with regard to the helical structure.
2. Line 46: meprins are also found in *Ciona intestinalis*, which is an urochordate and not a vertebrate; this has to be corrected
3. Lines 49-51: the authors should cite also latest publications; e.g. PMID: 34435182
4. Lines 68-69: the less defined specificity of meprin α compared to meprin β is only observed using a peptide library; cleavage of protein substrates also shows higher preference for Asp/Glu in P1'
5. Lines 81-84: meprin α and meprin β are both down-regulated in IBD (PMID: 30916990)
6. Line 92: additional reports should be cited (e.g. PMID: 33673639)
7. Fig. 2e: the mutations seem to be rather randomly distributed; is anything known from tumor patients carrying mutations at position Arg372?
8. Suppl. Fig. 3a: scale bar is missing
9. Suppl. Fig. 7a: It looks like degradation of procollagen rather than maturation; were the C-terminal prodomains cleaved off? What is the source of the collagen?
10. Suppl. Fig. 7c: Why is the gel for the C308A variant cropped?
11. Suppl. Fig. 9 is not correctly cited in the text.

Reviewer #3 (Remarks to the Author):

The authors present a beautiful study on the metalloprotease meprin- α forming a helical ultrastructure. It was a pleasure reading this manuscript. The results are surprising and exciting and well presented. The results built on earlier findings that Meprin α is the largest secreted protease known, up to 6 mDa in size, as visualised by electron microscopy of purified recombinant enzyme.

The results are highly significant to the fields of protease research, cell biology in general and structural biology in general.

I only have a few suggestions:

- could authors comment on stability or presence of such helical multimers under (patho-)physiological conditions; including the presence of a diverse "biochemical" environment such as the extracellular matrix, where further proteins might block/quench multimerization? The authors should perhaps do themselves the favour of not overinterpreting the mouse mouse urine findings after cisplatin treatment.

- The authors state that activation of the zymogen was accomplished by way of magnetic trypsin beads. Could authors comment on possible artefacts stemming from this? How were further tryptic cleavages avoided? Have authors determined n-terminal integrity of activated meprin alpha? Can this perhaps be deduced from the structural data?

- The authors very nicely combine their structural findings with biochemical activity data. Is there any reason to assume that the helical structure is involved in directing meprin alpha to "natural" substrates?

-The structural analysis appears to be of very high quality, as far as this reviewer can tell. Given that the authors highlight a particular "single particle like analysis": would it make sense to deposit details (i.e. software settings) of the bioinformatic workflow?

Reviewer #4 (Remarks to the Author):

In this manuscript, Bayly-Jones et al describe the structure of wild type meprin A by cryoEM and biochemical analysis to elucidate the role of high order oligomeric helical structure. Overall, the work offers important insight in the molecular basis how meprin A works. However, I have the following comments.

General comments:

- a. SAXS analysis to address structural feature of meprin A in solution: Authors should perform small angle x-ray scattering analysis of wild type and mutant of meprin A to substantiate the rod-shaped, helical nature of meprin A in solution.**
- b. Lengthy and not well focused introduction: The introduction is quite long and is not particularly focused. It should be better organized to convey key features relevant to this research and be shortened substantially.**
- c. Better terminology and method section to describe how structure was determined: Authors use "Single particle like cryoEM method" to describe how they solved meprin alpha mainly by using single particle cryoEM which is assisted by cryoET initial volume analysis. The approach seems to be different from the typical single particle cryoEM approach to macromolecules that are helical in nature, e.g., Ma et al Cell 179: 909, 2019. However, the term and method section are confusing. In addition, the main text should briefly mention how cryoET and cryoEM were used together to solve the structure. Some informative discussion how their method can be used in future analysis of helical structure of macromolecules will be useful if it is indeed very unique. If it is not unique, authors should point out the analogous studies.**
- d. Many descriptions in main text and methods are imprecise and sometimes confusing or incorrect: For example, it states in line 160 that "The meprin α oligomer resembles a spring of roughly 22 by 200 nm in dimensions, with some filaments observed to be greater than 500 nm long". This sentence does not indicate which one is for length and which is for "diameter?". If I understand correctly from figure, it may be stated as follows: The meprin α oligomer resembles a spring of roughly 22 nm in diameter and its length varies, 200 nm in average but can be greater than 500 nm. Another example is the statement line 196 that "frequently occurring cancer-associated mutations cluster to the MAM/MATH domains and, therefore, the interface of meprin α helices". However, based on the figure 2e, the mutations mostly are not located at interface. Authors should revise the manuscript throughout to make it precise.**
- e. Mismatch or misleading between text and figure: There are several discrepancies between text and figure. For example, line 184 - the dimer interface is most extensive (1374 A²) and highly conserved between both meprin α and meprin β ". However, the figure 2c shows multiple charge swap and charge loss mutations from local sequence alignment of MAM/MATH domains between meprin α and meprin β , which highlights the difference instead of conservation. Supplemental figure 4 only shows the overall sequence alignment, not specific for the interface sequence. Only supplemental figure 5 support this sentence. Authors should revise the manuscript throughout to avoid such discrepancy.**

Specific comments

- a. The role of meprin alpha in oncogenesis and driving aggressive metastasis is specifically highlighted in title and abstract. It is not clear that it is warranted.**
- b. Abbreviation for the structure domains of meprin alpha can be more logically presented and should be more uniform. For example, in line 96, protease domain (M12A) change to M12A (M12A clan protease) domain and in line 97, a MAM (meprin, A-**

- 5 protein, and receptor protein tyrosine phosphatase m) changes to a MAM (Meprin, A-5 protein, and receptor protein tyrosine phosphatase mu).
- c. Add "Result" to divide the result section from introduction, which is missing currently.
 - d. Supplemental figure b, denote the location and type of purification tag and secretion signal from insect. This is relevant because it is unclear how meprin alpha is biotin-tagged because it was purified using strep-tactin column while pMT/BiP/V5 was used as the expression vector in S2 cells where only V5 and hexahistidine should be present.
 - e. Line 173, the disulfide bond between MAN/MAN domains should be shown in figure 1e since it is discussed here.
 - f. Line 179, state the estimated size based on the molecular weight marker.
 - g. Need to provide information on the SEC experiments. The column/resin, in particular, would be useful as the elution profile in Fig. S1E which looks odd.
 - h. Explain what the black dots labeled "PC" in Fig. S1E refer to.
 - i. Explain why the non-reducing gel in Fig. S1D shows a (double?) band that are around 250-300 kDa. It should be much less.
 - j. Resolution of many figures are of poor quality. Figure S6 is particularly bad.
 - k. The authors should provide primer information for their site-directed mutagenesis.
 - l. The authors made several statements about side chain conformations and rotameric differences between structures. Given that these are EM structures that do not have particularly high resolution, authors need to display electron density observed for those residues to support their statements. Mostly in the last paragraph of the results.
 - m. Author should show examples of how the models fit the density, like how we show helices built into the density.
 - n. In line 533, "various amplitude corrected maps were employed for model building". It is unclear how it was done. If multiple maps were used for model building, they need to also be deposited. In addition, how the map-model fits and FSCs were calculated if multiple maps were used.
 - o. CryoET data should be deposited as well.
 - p. Meprin alpha/fetuin-B structure has a residue in the disallowed Ramachandran region. It should be fixed.
 - q. Table S1 their footnote needs to be improved. The Rosenthal and Henderson method should be cited. Their statement about sharpening "sometimes performed by deepEMhancer" is too vague. They need to be more specific about their methods.

REVIEWER COMMENTS

Reviewer #1 (Remarks to the Author):

“Helical ultrastructure of the oncogenic metalloprotease meprin α in complex with a small molecule hydroxamate inhibitor”.

Summary: The authors utilized cryo-electron microscopy to determine the high-resolution structure of the zymogen and mature form of meprin α , as well as the structure of the active form in complex with a prototype small molecule inhibitor and human fetuin-B. The authors also used mutagenesis to examine a role of meprin α oligomerization in its function and regulation and found that oligomerisation improves proteolytic and thermal stability but does not impact substrate specificity or enzymatic activity.

Comments: This is a very interesting, original, and important study for the meprin field. To the best of my knowledge, this is the first high-resolution structure of meprin α alone, and which is especially important, in the complex with the selective meprin α inhibitor 10d. The complex with meprin α inhibitor will enable further inhibitor/drug discovery studies, hopefully resulting in novel drugs.

The claims of proteolytic stability of meprin α oligomers and monomers are not sufficiently substantiated (see issues below). The rest of conclusions are well supported. No flaws in data analysis were observed. The methodology is sound, and the overall work is of high quality.

Issues:

Major:

1. Figure 3f. Trypsin treatment appears to produce smaller molecular size meprin α in both oligomeric and monomeric variants. C308A mutant which runs at 70kDa in both native and reducing conditions (Suppl fig 1C,D) runs at ~100kDa in Fig 3f. What is the reason for this inconsistency?

Qualitatively, the degradation products between trypsin and plasmin appear quite similar. Of course, trypsin and plasmin are different proteases. Therefore, we don't expect both proteases to yield identical degradation products.

Regarding the gel, we thank the reviewer for bringing this inconsistency to our attention. The molecular marker in 3f was incorrectly labelled. This has been fixed.

2. This reviewer agrees that at the time frame of proteolytic stability assay (20min) WT enzyme appears to be more stable. However, why the proteolytic digest was stopped after 20 min? Perhaps, if it was run longer there would be no or little difference between WT and mutants. Both C308A and R372T mutants appear to have digest products at ~50kDa after trypsin treatment and WT, C308A and R372T have ~50kDa products after plasmin treatment, which suggests that neither meprin α oligomers nor monomer are 100% proteolytically stable.

We agree with the reviewer that neither oligomers nor lower stoichiometric variants are 100% proteolytically stable. Indeed, this is not our conclusion. The manuscript states, “*In contrast however, dimeric meprin α was found to be less stable than the oligomeric form against proteolytic degradation by trypsin and plasmin during activation of the zymogens (Figure 3f)*”. Namely, the half-life of oligomers is longer than that of smaller stoichiometric variants.

We agree that stopping the reaction later would yield seemingly identical stability results. However, this would only be due to masking of genuine differences in proteolytic stability (over time). By choosing an appropriate time (e.g. 20 mins), differences in the decay rate can be observed. The hypothetical graph below illustrates how an “end-point assay” can be misleading, or arbitrarily chosen to yield identical results (e.g. >10 min).

We believe a better experiment would be a “time-resolved” assay, however we currently do not have a means of performing this measurement.

Minor:

1. The conclusions section would be helpful.

For brevity, the conclusions are merged within the discussion section.

2. Which tight-binding inhibitor of meprin α was utilized for determination of active enzyme? To the best of our knowledge, there are no tight-binding inhibitors of meprin α in the literature.

The reviewer is right. The inhibitor is not published yet but publication (by Tan and Ramsbeck) is in preparation and will be finished soon. K_i app determined by Morrison equation is 150 pM.

3. The authors mention that off-target inhibition of proteolytic activity is a current limitation of existing candidates. However, Wang 2021 et al., paper (PMID: 33673639) lists meprin α inhibitor with more than 100-fold selectivity vs meprin β and related zinc-metalloproteases.

We agree with the reviewer, the R26466 compound described by Wang et al. and Hou et al., is indeed highly selective (125-fold). We note the potency of this compound is weaker (~ 600 nM) compared to e.g., compound 10d (IC₅₀ 160 nM; 18-fold selective) or SR162808 (IC₅₀ 300 nM; 38-fold selective) (Hou et al., 2021).

We have rephrased the manuscript to highlight this nuance where potency and selectivity reflect a current trade-off for inhibitor candidates. This being the general point we wished to make – that it remains difficult to find highly potent and highly specific compounds.

We have also cited these publications as they're clearly relevant to the discussion.

3. Figure 3d. IC₅₀ values are not constants as they are heavily dependent on the assay conditions, but rather estimates of true potency. The preferred terminology for IC₅₀ values is “an apparent potency”.

We have changed the terminology to “apparent potency” where applicable.

4. It is not entirely clear whether the meprin α structure was solved in complex with 10d and fetuin or ligands were docked into the meprin α structure. Different claims are made throughout the paper, therefore, it needs to be better explained.

In fact, complexes of meprin (zymogen form, active form, complex with 10d and complex with fetuin-B) were all determined by cryo-EM. We have deposited all these maps and models to the EMDB and PDB respectively.

In the context of fetuin-B, computation docking analysis was performed initially and subsequently we conducted cryoEM experiments to validate our docking analysis.

“We therefore imaged meprin α in complex with human fetuin-B by cryo-EM and generated a 3.7 Å resolution reconstruction (Supp Figure 8g). Indeed, this structure confirms the intricate packing suggested by the molecular docking analysis, albeit with some degree of flexibility.”

We are unsure which statements in the manuscript have given conflicting impressions regarding compound 10d. We cite the relevant sentences below:

“To characterise the structural basis of binding and mechanism of inhibition by the small molecule inhibitor, we determined the structure of meprin α after co-incubation with compound 10d.”

“Similarly, compound 10d was added in excess (~8 fold K_i) and allowed to incubate for 1 minute, prior to freezing.”

“The cryoEM density of compound 10d is shown in green, which is deeply buried within the cleft.”

“Chemical docking simulations were used to flexibly fit the ligand structure into the density.”

In the model building section we describe the method of structural modelling, where atomic models of the ligands were built by initially “docking” crystal structures or AlphaFold models into the cryoEM density prior to then refining these models. We have slightly modified the methods to further clarify.

“Subsequently, a model of the meprin α /fetuin-B complex was built into the cryoEM density. Initially, we took the model of active meprin α and the AlphaFold model of human fetuin-B^{98,99}. These were rigid body fit into the cryo-EM reconstruction and were subsequently flexibly fit and interactively refined in ISOLDE/ChimeraX using secondary structure and all atom constraints to prevent the model from diverging.”

Reviewer #2 (Remarks to the Author):

In this manuscript, the authors report on the high-resolution quaternary structure of the metalloprotease meprin α , resolved by cryo-electron microscopy.

Meprin α has been reported in numerous organs and cell types like the intestine, the kidney, the skin and different leukocyte populations. Dysregulated expression, localization and activity of meprin α has been shown to be implicated in pathophysiological processes of various inflammatory diseases like inflammatory bowel diseases (IBD), acute kidney injury, Kawasaki syndrome (a systemic vasculitis of unknown etiology in childhood) and sepsis. The substrate pool of meprin α comprises numerous inflammatory mediators including cytokines, chemokines, extracellular matrix molecules, immune-regulatory adhesion molecules and receptors. Therefore, meprin α has been investigated as putative pharmacological target, and this requires structural information for proper drug design. Previous studies analyzing recombinant and purified meprin α indicated already huge meprin α homooligomers, determined by analytical gel filtration and transmission electron microscopy, up to 6 MDa in size.

However, the real molecular assembly and identification of interaction sites for oligomerization were ambiguous. Unlike the close relative meprin β , which is exclusively forming dimers that allowed for resolving the crystal structure of the protease, identification of the oligomeric structure of meprin α requires different strategies. Herein, the authors revealed that meprin α builds a flexible, left-handed helical assembly, with a rather stringent diameter of about 22 nm but various length up to 500 nm, which represents a 27 MDa complex. This makes meprin α by far the largest secreted proteolytic enzyme. Additionally, the authors generated several variants of meprin α that nicely confirmed the amino acid residues responsible for oligomerisation (particularly Arg372) and furthermore provided structural data of meprin α in complex with the naturally occurring inhibitor fetuin B and a small synthetic molecular compound.

This paper merits publication in the journal Nature Communications, because it reports on the structural properties of the largest extracellular protease meprin α , building a fascinating helical structure that provides locally focused massive proteolytic activity, which certainly has specific biological function. This manuscript is the basis for many follow-up stories that aim to unravel these functions.

We thank the reviewer for their positive reception of our work. We agree that many follow-up studies are now necessary. We hope our results can define a foundation to conduct that work. We look forward to the contributions of our colleagues in complementary fields, such as cellular biology and animal work, and would welcome collaboration in these efforts.

However, there are several points that need to be addressed prior publication.

Major issues:

1. For the experiments in Fig. 3f and Suppl. Figures 1e, 6a-c, 7 the meprin α variant R372A should be included. This is the perfect control, as it solely builds dimers.

We thank the reviewer for their remark. We have now included additional experiments outlined below.

In the process of our revisions, we discovered contradictory results between mass photometry and the original cryoEM imaging. We therefore repeated the cryoEM imaging of all variants. We found that at high concentration ($\sim\mu\text{M}$) that R372A and R372T are *both* capable of forming small oligomers, consistent with 8- or 12-mers. We believe these were not observed in the original cryoEM images of R372A, as very thin ice occludes larger particles and forces these out of the imaging field of view. In our repeated experiments, we specifically imaged thicker ice regions where we found the presence of oligomers. We have updated the manuscript accordingly.

Conversely at nanomolar ranges, the equilibrium of dimer/oligomer formation for R372T appears to be weaker than R372A (for which we observe a population of tetramers). Ultimately these data suggest that the oligomerisation of meprin α (at least with respect to R372T) is somewhat concentration dependent. Notably, very large oligomers were still present at 10 nM when we conducted mass photometry measurements of wild-type material.

Due to limited expression yields, we were unable to perform full characterisation of the R372A variant. However, it appears that R372T is perhaps a more suitable control and fortunately this variant expresses well. We have updated the following:

Fig 3f – comparison between WT (filamentous), C308A (polydisperse) and R372T (only dimer at low concentration)

Supp Fig 6 – new addition of MALDI-TOF for C308, and addition of mass photometry (including R372A).

“b. Mass photometry single-molecule mass distribution of meprin α , β and variants.”

We have resultantly updated our discussion in the text surrounding the oligomers – see “Analysis of meprin oligomerisation”.

Supp Fig 7 – Substrate degradation comparison between WT, C308A, and R372T. Unfortunately, fibronectin was not available and we were unable to perform substrate degradation on this sample.

2. For Suppl. Fig. 1e standard proteins should additionally be loaded to better evaluate the different oligomeric states of meprin α .

We have modified the figure to include the standard protein calibration for clarity. Dot immunoblots were not performed on protein standards.

We feel that it is important to emphasise that size exclusion is not an accurate method for mass determination. We simply wished to illustrate similarity between murine native meprin α and recombinant meprin α . We have conducted a more sophisticated analysis of mass/stoichiometries in Supp Fig 6. We are sure the reviewer appreciates that mouse urine is a very heterogeneous sample and is not amenable to mass photometry or mass spectrometry, which is why we had chosen to do size exclusion analysis and immunoblot.

3. Lines 228-229: The authors stated that the meprin α variant C308A is not capable of forming dimers anymore, due to the loss of the stabilizing disulphide bridge. Here, the authors should explain in more detail, how the oligomers seen in Suppl. Fig. 6d can assemble.

It seems that C308 mostly has a stabilising effect, where loss of the disulphide bond results in an equilibrium driven by weak self-association. Thus, the helical interface and the central interface together drive assembly as in wild type meprin α , however the weakened interface is prone to dissociation. We speculate that a critical number of subunits can assemble, before the size and diffusion result in significant sheer forces that favour dissociation, thereby placing an upper limit on the filament size for the C308A variant.

Both mass photometry and cryoEM imaging indicate that C308A produces dimers, tetramers, hexamers, and higher order oligomers. We have added mass photometry to further support these claims.

“b. Mass photometry single-molecule mass distribution of meprin α , β and variants. The mass distribution of wild type meprin α appears skewed toward smaller oligomeric species, however this is artificial since the largest filaments are excluded during event detection. Nevertheless, the highly polydisperse nature of the wild-type sample is apparent.”

“Single-molecule mass measurements by mass photometry revealed the covalent and non-covalent interfaces were disrupted by mutagenesis to different extents yielding mixed populations of different stoichiometries (Supp Figure 6b). At low concentrations (10 nM) complete dissociation of the helical interface was observed for R372T, while R372A maintained weak association to assemble tetramers. Similarly, C308A maintained the ability to associate into a range of stoichiometries, however very large oligomers were unfavoured illustrating the stabilising role of the C308 disulphide bridge. Unlike the mutant variants, very large oligomers (well above the detection limit of mass photometry software) were observed in the single-molecule movies.”

In this regard, it obviously happened that the spectra for meprin α C308A displayed in Suppl. Fig. 6a were cropped. Please show the entire graphs.

We have repeated the MALDI-TOF measurements and amended the figure to include the full range of m/z values.

4. Since only one amino acid (Arg372) is crucial for the ionic interaction of the head-to-tail oligomer, the authors should provide data in how far salt concentrations and pH differences influence stability and size of the helical structure. For instance, the detected size of meprin α in urine appears to be smaller than the size of rec. enzyme (Suppl. Fig. 1e).

We note the comparison between recombinant meprin and that obtained from urine may be due to differences between the human and murine homologs. Nevertheless, we do see a population of murine meprin α in urine that is consistent with recombinantly produced enzyme (around 8-10 mL). However, the salt or pH may also be a contributing factor (see below).

“Consistent with this observation, cryoEM imaging of meprin α oligomers in the presence of high salt or acidic conditions resulted in disruption of the helical interface producing smaller oligomeric species (Supp Figure 7). Conversely, alkaline conditions preserve the oligomeric state.”

It was not possible to measure the full filamentous oligomeric sample in the presence of different salts and pH by SEC owing to the amount of required enzyme and difficulties owing to its size. Instead, we conducted cryoEM imaging in high salt conditions and high or low pH, indeed as expected high salt conditions appear to destabilise the oligomeric form (compared to 100 mM NaCl at pH 9 or 7.4). Similarly, in low pH conditions (pH 5) we were unable to see the normal oligomeric filaments indicating the electrostatic interactions between the MAM/MATH domains were once again disrupted.

5. Can the authors predict a model, if helical oligomerisation of meprin α could theoretically also start from a meprin α /meprin β heterodimer at the cell surface? That would consequently result in massive local meprin α activity at the plasma membrane.

In principle, yes this is possible, however it remains to be shown experimentally. A single meprin α /meprin β heterodimer may nucleate the growth of a larger meprin α oligomer, since the meprin α helical interface is available in meprin α /meprin β heterodimer. However, the formation of heterotetramers may be favoured due to their localisation on the membrane, and this would prevent the nucleation of larger oligomers.

Regarding the local meprin α activity, this may still occur due to the limited diffusion of meprin α filaments. These effects are yet to be shown, but we believe they are an interesting implication of our results.

6. The authors stated that the identified helical interface in meprin α is missing in meprin β . The alignment in Fig. 2c should be extended for additional species. It would be interesting to see, whether the capability of meprin α to form huge oligomers is preserved throughout evolution of the enzyme, which would underline the likely biological impact.

Yes, we agree – the conservation of the helical interface across species (between orthologs) supports a biological role for the meprin α oligomer with an evolutionary purpose.

Supp figure 5 shows the surface conservation of the helical interface derived from sequence alignments of numerous (>100) meprin α orthologs. This is a 3D representation of the multiple sequence alignment in the context of the interface. We believe the surface conservation is more informative than a standard sequence alignment, as one can visually see the clustering of discontinuous conserved regions to the interface.

7. Fig. 4 and Suppl. Fig. 10: The part concerning the small molecule inhibitor is not fully convincing to me. Is the orientation of the small compound really fully resolved in the cryo-EM structure? The explanation for the specificity towards meprin α over meprin β should be further defined. From the picture shown in Suppl. Fig. 10 the Arg238 in meprin β may adopt several other rotamer positions that would not interfere with the inhibitor.

Yes, we resolve the small molecule ligand to 2.4 Å. We show the cryoEM density for the small molecule ligand in Supp Fig 10 (in green). Since the resolution is 2.4 Å, we cannot resolve water networks and the central cavity of aromatic rings is not yet resolved.

We have included four new panels in Supp. Fig. 10 illustrating model and map fit of Arg242 (Arg238 meprin β). In the original meprin β (PDB 4GWN) structure, the position of R238 is somewhat ambiguous in our opinion (although it is modelled), however in a slightly higher resolution structure (PDB 7AQ1) the rotamer is quite clearly different from meprin α .

We have also rephrased the last paragraph for clarity.

“The only exceptions being I152 (T149 in meprin β) and Q215 (S212 in meprin β). As a result, it remains unclear at the current resolution how compound 10d derives its selectivity for meprin α .”

Additionally, ovastacin, another astacin metalloprotease, is also inhibited by fetuin B. Does compound 10d inhibit this protease too, and if not, what are the molecular clashes that prevent the binding.

According to Tan et al. (2021), compound 10d has a K_i^{app} of $1.15 \pm 0.07 \mu\text{M}$ for ovastacin. This indicates higher selectivity for meprin α by 10d with a K_i^{app} of 52.25 nM.

Currently there is no experimentally determined structure of the zymogen or active forms of ovastacin. However, the AlphaFold model (AF-Q6HA08-F1; right) predicts the same rotamer position of R264 for ovastacin as R238 in meprin β (PDB-7AQ1).

Once again, the active site is remarkably similar, with Y145 (meprin α) being substituted for a lysine in ovastacin (K175) – see also comment above. Ultimately these predictions may be inaccurate – so it will be necessary to obtain crystal or EM structures of ovastacin.

Minor issues:

1. The title could be edited with a stronger focus on the unique structural properties of meprin α . The ‘oncogenic’ function of meprin α is not experimentally addressed in this manuscript and other biological activities, e.g. in the brush borders of the intestine and kidney, might be more relevant with regard to the helical structure.

We have modified the title:

“Helical ultrastructure of the metalloprotease meprin α in complex with a small molecule inhibitor”

2. Line 46: meprins are also found in *Ciona intestinalis*, which is an urochordate and not a vertebrate; this has to be corrected

We are aware of two conflicting reports, according to Huxley-Jones et al. [1], it is stated “Meprin orthologues were not found in the *ciona* genome.” However, more recently, Ignacio Marín [2] claims “three genes encoding structurally typical meprin proteins are arranged in tandem in the genome of the urochordate *Ciona intestinalis*”. We have modified the text accordingly.

“Meprin α , together with the evolutionary related meprin β , represent a subgroup of astacins⁹.”

[1] “The evolution of the vertebrate metzincins; insights from *Ciona intestinalis* and *Danio rerio*”. Julie Huxley-Jones¹, Toni-Kim Clarke^{1,2}, Christine Beck¹, George Toubaris¹, David L Robertson¹ and Raymond P Boot-Handford¹

[2] Marín I (2015) Origin and Diversification of Meprin Proteases. PLOS ONE 10(8): e0135924. <https://doi.org/10.1371/journal.pone.0135924>

3. Lines 49-51: the authors should cite also latest publications; e.g. PMID: 34435182

We have included the suggested reference.

“Differential regulation and localisation of meprins provides access to these substrates, enabling specific functions in pro- and anti-inflammatory responses, ECM assembly and remodelling, wound healing, cytokine activation and signalling, as well as cell-cell adhesion¹⁹⁻²⁷.”

4. Lines 68-69: the less defined specificity of meprin α compared to meprin β is only observed using a peptide library; cleavage of protein substrates also shows higher preference for Asp/Glu in P1’

We have updated the text accordingly:

“To achieve their function, both meprins cleave a specific but distinct motif. Both meprin α and β are characterised by a striking preference for negatively charged amino acids in the P1’ position unlike most other extracellular proteases³⁴. Nevertheless, meprin α and β partially discriminate between substrates, resulting in distinct activity profiles.”

5. Lines 81-84: meprin α and meprin β are both down-regulated in IBD (PMID: 30916990).

We wished to convey that specific inhibition of meprin α is desirable in contexts where meprin α drives diseases, but where meprin β remains beneficial.

We have modified the paragraph to make this point more clearly.

“Importantly dysregulation of meprin α can occur in contexts where meprin β retains normal regulatory and physiological functions. As such, modulation of meprin activities may have either beneficial or deleterious effects on health depending on the context. For example, meprin β shows a protective effect in IBD and promotes mucus turnover by cleavage of MUC2, preventing bacterial overgrowth in intestinal mucosa^{32,39,40}.

In general, specific inhibitors for meprin α and meprin β separately might be useful to treat certain diseases, such as progressive cancers, without disrupting the physiological function of the other homologue. For these reasons, specific inhibitors of meprin α have been sought to mitigate its function in disease progression and to investigate the pathophysiological role of meprins in further detail.”

6. Line 92: additional reports should be cited (e.g. PMID: 33673639)

We have included both Wang et al. (2021), and Hou et al. (2021).

7. Fig. 2e: the mutations seem to be rather randomly distributed; is anything known from tumor patients carrying mutations at position Arg372?

We were unable to find any reports that mention R372. Indeed, many other mutations are present in meprin α , which appear to be throughout the protein (Extended Data 1). However, the most frequently mutated residues cluster to within the MAM and MATH domains.

8. Suppl. Fig. 3a: scale bar is missing

Thanks, we have fixed this.

9. Suppl. Fig. 7a: It looks like degradation of procollagen rather than maturation; were the C-terminal prodomains cleaved off? What is the source of the collagen?

Supp 7a - depicted is digestion of mature Collagen I prepared from rat tail (according to Gorisse) as described in the Method section.

10. Suppl. Fig. 7c: Why is the gel for the C308A variant cropped?

This was the result of a loading error requiring the gel to be cropped to maintain the correct order. We have now replaced the figure in question with a repeated experiment, according to our response to your first remark.

11. Suppl. Fig. 9 is not correctly cited in the text.

Thanks – we have fixed this error.

Reviewer #3 (Remarks to the Author):

The authors present a beautiful study on the metalloprotease meprin-alpha forming a helical ultrastructure. It was a pleasure reading this manuscript. The results are surprising and exciting and well presented. The results built on earlier findings that Meprin α is the largest secreted protease known, up to 6 mDa in size, as visualised by electron microscopy of purified recombinant enzyme.

The results are highly significant to the fields of protease research, cell biology in general and structural biology in general.

We thank the reviewer for their time and well-received suggestions.

I only have a few suggestions:

- could authors comment on stability or presence of such helical multimers under (patho-)physiological conditions; including the presence of a diverse “biochemical” environment such as the extracellular matrix, where further proteins might block/quench multimerization? The authors should perhaps do themselves the favour of not overinterpreting the mouse urine findings after cisplatin treatment.

In general, the presence of helical multimers in vivo needs to be investigated in further studies. Their size may depend on several factors like concentration, molecular crowding effects, pH, salt, and others. We do not expect the helical oligomers to be as large as those observed in vitro from recombinant material (i.e., up to 2 μm !). We agree with the intuition of the reviewer, that crowding, concentration and other factors will likely limit the growth of meprin filaments.

Nevertheless, we anticipate large oligomers do form (perhaps single digit megadalton, ~100 nm). Our results define how meprin multimers are organized, their symmetry, how the interaction is defined, etc.

We also agree with the reviewer that the data obtained with mouse urine should not be over-interpreted. Please see our comment to reviewer #2, remark 2. These experiments were simply to compare recombinant and native meprin, to assess whether oligomerisation was purely an artefact of our construct (we don't believe so) or whether we could observe oligomeric meprin from native source material.

We have also slightly rephrased our statement:

“To determine whether higher-order oligomers could also be found from native source material, we conducted an analytical size-exclusion chromatography analysis of mouse urine after cisplatin treatment. Indeed, we observed evidence that native meprin α is capable of forming oligomers of similar molecular mass, albeit the major population appeared smaller than recombinant wild type meprin α (Supp Figure 1e).”

- The authors state that activation of the zymogen was accomplished by way of magnetic trypsin beads. Could authors comment on possible artefacts stemming from this? How were further tryptic cleavages avoided? Have authors determined n-terminal integrity of activated meprin alpha? Can this perhaps be deduced from the structural data?

The activation with magnetic trypsin beads was performed on one hand to get rid of the trypsin without further purification steps of meprin (as we experienced a dramatic loss of protein when performing SEC or ion exchange chromatography obviously due to the huge size of the multimers that are not entering the stationary phase) and on the other hand to reduce tryptic digestion due to steric hindrance of immobilized trypsin.

This was a major problem when activating the mutant meprin with soluble trypsin. Activation was accompanied by simultaneous loss of activity due to further degradation. This could be prevented to some extent by using the immobilized trypsin although not completely. Therefore, the active molecules of the proteins were determined by titration with a tight binding inhibitor for calculation of kinetic constants.

N-terminal integrity was not determined chemically however as the reviewer suggests, we could resolve the N-terminus in our cryoEM reconstructions of wild-type meprin α , which appeared consistently between experiments. Owing to the averaging of many thousands of molecules, we do not see evidence of additional cleavage events in terms of broken density.

- The authors very nicely combine their structural findings with biochemical activity data. Is there any reason to assume that the helical structure is involved in directing meprin alpha to “natural” substrates?

We currently do not have experimental evidence to confidently claim that the ultrastructure is involved in scaffolding or directing interactions to natural substrates. But this is an appealing hypothesis, ultimately this needs to be addressed in further studies. Currently, the oligomeric nature does not appear important for substrate recognition or activity, at least in vitro.

An alternative hypothesis is that in the context of the extracellular matrix, owing to the highly crowded environment, oligomerisation limits the diffusion of meprin and thereby regulates local activity. However, this remains to be shown. For the sake of completeness and given that it is still unclear, we have added the reviewer's above comment as a discussion point:

“An alternative hypothesis is that the oligomeric ultrastructure somehow mediates interactions with certain substrates or has a scaffolding role.”

-The structural analysis appears to of very high quality, as far as this reviewer can tell. Given that the authors highlight a particular “single particle like analysis”: would it make sense to deposit details (i.e. software settings) of the bioinformatic workflow?

We have modified the methods section according to a request from reviewer #4. We have tried to provide as much detail as possible in the method section.

Reviewer #4 (Remarks to the Author):

In this manuscript, Bayly-Jones et al describe the structure of wild type meprin A by cryoEM and biochemical analysis to elucidate the role of high order oligomeric helical structure. Overall, the work offers important insight in the molecular basis how meprin A works. However, I have the following comments.

General comments:

a. SAXS analysis to address structural feature of meprin A in solution: Authors should perform small angle x-ray scattering analysis of wild type and mutant of meprin A to substantiate the rod-shaped, helical nature of meprin A in solution.

We provide a combination of MALDS (in-solution), mass spectrometry (in vacuum) and cryoEM imaging (in-solution; achieving sub-3 Å resolution). However, to address other issues we have additionally conducted mass photometry analysis which further demonstrates the highly polydisperse nature of meprin α (in-solution) corroborating the above analyses.

We believe these data already strongly substantiate the filamentous nature of meprin α and that SAXS would be an unnecessary additional demonstration. Furthermore, the filamentous nature of meprin has also been reported by two other independent groups (Bertenshaw, 2003; Becker-Pauly, 2003).

b. Lengthy and not well focused introduction: The introduction is quite long and is not particularly focused. It should be better organized to convey key features relevant to this research and be shortened substantially.

Respectfully, we believe this is a matter of personal preference and does not substantially affect the scientific content or conclusions. Given that there appears to be a consensus among the four reviewers, in this instance we have opted to leave the introduction unchanged.

c. Better terminology and method section to describe how structure was determined: Authors use “Single particle like cryoEM method” to describe how they solved meprin alpha mainly by using single particle cryoEM which is assisted by cryoET initial volume analysis. The approach seems to be different from the typical single particle cryoEM approach to macromolecules that are helical in nature, e.g., Ma et al Cell 179: 909, 2019. However, the term and method section are confusing. In addition, the main text should briefly mention how cryoET and cryoEM were used together to solve the structure. Some informative discussion how their method can be used in future analysis of helical structure of macromolecules will be useful if it is indeed very unique. If it is not unique, authors should point out the analogous studies.

A cryoET/STA approach was initially tried when we attempted to solve the structure of meprin α in 2018. This yielded low resolution reconstructions with roughly 7 nm resolution (the spacing of the helix turns). We subsequently used these initial volumes to bootstrap the SPA reconstructions and guide our analysis. The resulting reconstructions from SPA were also limited in resolution. Ultimately, localised reconstructions were necessary to overcome the hard resolution barrier (~10-12 Å) imposed by flexibility. In retrospect, the use of STA was likely not necessary, as now ab initio methods yield reconstructions of similar quality that are sufficient for SPA. We have updated our STA methods and maps, these are deposited and also included in the manuscript. The STA similarly was resolution limited to 10-12 Å.

We chose to describe the analysis as “single-particle like” or “pseudoSPA” since the reconstructed object is not a true independent isolated particle. That is, the assumption of identical randomly oriented isolated particles is not satisfied (it’s not really “single-particle”) – this is the case for most helical refinements. To our knowledge, the approach has similar elements to that described for bacterial flagella (Shibata et al., 2019; Kato et al, 2019). Isolating sub-particles directly at the stage of picking did not work, initial guesses at the 3D alignments were necessary.

We have made substantial changes to the methods and main text to make this clearer.

d. Many descriptions in main text and methods are imprecise and sometimes confusing or incorrect: For example, it states in line 160 that “The meprin α oligomer resembles a spring of roughly 22 by 200 nm in dimensions, with some filaments observed to be greater than 500 nm long”. This sentence does not indicate which one is for length and which is for “diameter?”. If I understand correctly from figure, it may be stated as follows: The meprin α oligomer resembles a spring of roughly 22 nm in diameter and its length varies, 200 nm in average but can be greater than 500 nm.

We have rephrased the sentence based on the reviewer’s recommendation.

“The meprin α oligomer resembles a spring of roughly 22 nm in diameter and 200 nm in length (Figure 1a, Supp Fig 3), with some filaments observed to be greater than 500 nm long (~27 MDa).”

Another example is the statement line 196 that “frequently occurring cancer-associated mutations cluster to the MAM/MATH domains and, therefore, the interface of meprin α helices”. However, based on the figure 2e, the mutations mostly are not located at interface. Authors should revise the manuscript throughout to make it precise.

We have rephrased this sentence as follows:

“In this regard, we note that frequently occurring cancer-associated mutations cluster to the MAM/MATH domains. Given that these domains define the interface of meprin α helices, mutations that affect the fold of the MAM or MATH domains would resultantly affect interface packing (Figure 2e; Extended Data 1)⁵⁸. In contrast, meprin β cancer-associated mutations more frequently cluster within in the M12A protease domain (Extended Data 1)⁵⁸.”

e. Mismatch or misleading between text and figure: There are several discrepancies between text and figure. For example, line 184 - the dimer interface is most extensive (1374 Å²) and highly conserved between both meprin α and meprin β ”. However, the figure 2c shows multiple charge swap and charge loss mutations from local sequence alignment of MAM/MATH domains between meprin α and meprin β , which highlights the difference instead of conservation. Supplemental figure 4 only shows the overall sequence alignment, not specific for the interface sequence. Only supplemental figure 5 support this sentence. Authors should revise the manuscript throughout to avoid such discrepancy.

Thanks for picking up this confusing reference. Since the dimer interface and helical interface are not equivalent, we refer to two different interfaces in our analysis. Obviously here it was not clear and we have re-arranged the figure.

Figure 2d (previously 2c) shows differences in conservation of the helical interface (which is understandably different between meprin α and β), while line 184 refers to the central homodimer interface (which is conserved between the orthologs and homologs – shown in Fig S5 and S6).

Specific comments

a. The role of meprin alpha in oncogenesis and driving aggressive metastasis is specifically highlighted in title and abstract. It is not clear that it is warranted.

We have removed the term “oncogenic” from the title and have modified the abstract. The role of meprin in metastasis is a key point of interest and therefore we have included this in the discussion and introduction.

Title: “Helical ultrastructure of the metalloprotease meprin α in complex with a small molecule inhibitor”

Abstract: “Dysregulated meprin α is associated with acute kidney injury, sepsis, urinary tract infection, metastatic colorectal carcinoma, and inflammatory bowel disease.”

b. Abbreviation for the structure domains of meprin alpha can be more logically presented and should be more uniform. For example, in line 96, protease domain (M12A) change to M12A (M12A clan protease) domain and in line 97, a MAM (meprin, A-5 protein, and receptor protein tyrosine phosphatase m) changes to a MAM (Meprin, A-5 protein, and receptor protein tyrosine phosphatase mu).

We have made the suggested changes.

c. Add “Result” to divide the result section from introduction, which is missing currently.

We have added the section title.

d. Supplemental figure b, denote the location and type of purification tag and secretion signal from insect. This is relevant because it is unclear how meprin alpha is biotin-tagged because it was purified using strep-tactin column while pMT/BiP/V5 was used as the expression vector in S2 cells where only V5 and hexahistidine should be present.

Strep-tactin columns also bind a peptide epitope tag known as “Strep-tag”, whose sequence can be genetically encoded as “WSHPQFEK”. As such, we do not (nor do we need to) biotinylate meprin. The strep-tag was introduced using a specific sense primer for amplification.

Meprin α was cloned into pMT/BiP/V5-HisA using NcoI and NotI, and as a result is directed for secretion by the BiP secretion peptide (present in the vector). The native meprin α secretion signal was omitted. A stop codon was introduced after residue 600 in meprin alpha and as such the C-terminal V5 and hexahistidine tags are not expressed.

We have modified the method section and supplemental figure b for clarity to show the construct specific modifications.

“Briefly, the sequence of pro-meprin α (V22-S600, i.e. omitting the native signal peptide; uniprot Q16819) containing an N-terminal Strep-Tag (WSHPQFEK) was cloned into pMT/BiP/V5 with a C-terminus stop codon after residue S600, enabling stable cell lines to be produced. The pMT/BiP/V5 vector encodes the BiP signal peptide enabling meprin secretion with an N-terminal Strep-Tag for purification.”

e. Line 173, the disulfide bond between MAN/MAN domains should be shown in figure 1e since it is discussed here.

We have modified the figure to show the disulfide bond. A more detailed analysis is present in the next section of the manuscript, where we focus on the interfaces.

“... while at the dimer interface a conserved disulphide bond stabilises MAM/MAM interactions (see next section).”

f. Line 179, state the estimated size based on the molecular weight marker.

We perform a more comprehensive analysis of the mass and stoichiometry in a later section. We have added the following sentence:

“The apparent molecular mass based on SDS-PAGE was observed to be 70-73 kDa, which is notably higher than the theoretical mass of 67.7 kDa, most likely due to glycosylation (Supp Figure 1c, d).”

g. Need to provide information on the SEC experiments. The column/resin, in particular, would be useful as the elution profile in Fig. S1E which looks odd.

We have included column details in the figure legend. The column resin is Superose6. The elution profile is coarsely defined (which is why it appears “odd”). Here, each fraction was quantified by dot immunoblot, the trace shows immunoblot intensity and not by UV-Vis absorbance. This was done since urine has other proteins present and absorbance is not a direct measure of meprin.

h. Explain what the black dots labeled “PC” in Fig. S1E refer to.

We have updated the figure legend. “PC” refers to “pre-column” material that was injected into the instrument.

“PC: injected material (i.e. pre-column) applied as a control.”

i. Explain why the non-reducing gel in Fig. S1D shows a (double?) band that are around 250-300 kDa. It should be much less.

In non-reducing conditions, the covalent dimer will have a theoretical molecular mass of ~150 kDa. The reviewer is correct in pointing out the discrepancy between the apparent and actual molecular masses. With respect to meprin, we often see this phenomenon and we are aware that it is a common artefact of SDS-PAGE or electrophoresis in general. Indeed, Bertenshaw (2003) also observe this effect for recombinant rat meprins. This is clearly related to the disulphide bond, since in reducing conditions, the discrepancy vanishes.

The double band may be due to different glycosylation states.

j. Resolution of many figures are of poor quality. Figure S6 is particularly bad.

This may be a result of the manuscript tracking system and compiled PDF. High resolution figures are, of course, available and will be provided during typesetting. We have also verified upon re-submission that the quality is sufficient and that the figures are legible. Figure S8 has been modified for clarity (see below).

k. The authors should provide primer information for their site-directed mutagenesis.

We have included a new supplementary table to include the primers.

pMTB MepA_Strep (NcoI)	5'-ATACCATGGTGGTCCCACCCCCAGTTGAGAAGGTACCGATTAAGTATCTTC-3'
pMTB MepA_S600 (NotI)	5'-TATATACGCCGGCGAATCGACTCCACCCACTATAG-3'
hMepA_C308A_f	5'-ACACCTTGTTGGGACAAGCCACAGGTGCCGGCTACTTC-3'
hMepA_C308A_r	5'-GAAGTAGCCGGCACCTGTGGCTTGTCCCAACAAGGTGT-3'
hMepA_R372T_f	5'-AGCACAGGCAATGTTACCAAGTTGGTGAAGGTG-3'
hMepA_R372T_r	5'-CACCTTCACCAACTTGGTAACATTGCCTGTGCT-3'
hMepA_R372A_f	5'-AGCACAGGCAATGTTGCCAAGTTGGTGAAGGTG-3'
hMepA_R372A_r	5'-CACCTTCACCAACTTGGCAACATTGCCTGTGCT-3'

l. The authors made several statements about side chain conformations and rotameric differences between structures. Given that these are EM structures that do not have particularly high resolution, authors need to display electron density observed for those residues to support their statements. Mostly in the last paragraph of the results.

Please see response to reviewer #2, remark 7. We have now added figure panels showing the rotamer positions.

m. Author should show examples of how the models fit the density, like how we show helices built into the density.

We have included three selected regions of map to model agreement in Figure S3. The reader is also encouraged to download each map and model to inspect the agreement.

“e. Example regions of map and model agreement of the 2.4 Å meprin a reconstruction. Further details are available from the PDB and EMDB depositions (see Data availability).”

n. In line 533, “various amplitude corrected maps were employed for model building”. It is unclear how it was done.

This is common practice. For example, Coot now provides options to dynamically amplitude sharpen a map while modelling so the user can inspect different regions in a map with varying local resolution. Similar to changing the sigma threshold. We feel depositing all variations of sharpened maps is impractical and also not informative for the reader.

A single sharpened map is not representative of the data. Multiple sharpening analyses with different *B*-factors or indeed different methods (e.g. LocScale, bLocFilt, deepEMhancer, phenix.auto_sharpen, phenix.resolve_cryo_em, etc.) generate maps that may be more interpretable in some regions where other maps may be over or under sharpened. Toggling between these maps can be helpful guides while modelling.

If multiple maps were used for model building, they need to also be deposited.

In an effort to ensure our modelling and maps can be independently inspected and verified, we have deposited the independent half maps, full map, refinement mask and in some cases up to three sharpened maps for each of our seven

deposited EMDB entries. The inclusion of half maps and the full map means anyone can perform their own independent validation (local resolution, FSC calculations, anisotropy calculations, map-to-model FSC) or sharpening experiments, if they desire.

In addition, how the map-model fits and FSCs were calculated if multiple maps were used.

FSCs were calculated between independent half maps only and this is not dependent on map sharpening. Regarding map-model fits, these were calculated against a single (best) sharpened map.

o. CryoET data should be deposited as well.

Ultimately the STA reconstruction was not considered for the analyses of this paper and was therefore omitted. An aggressively low pass filtered sub-tomogram volume (not an average) from a single tomogram was used as the initial volume.

We have taken this opportunity to re-analyse the data within relion-4 and have achieved an improved sub-tomogram average with nominal resolution of 12.7 Å (see new supp figure 2). It is deposited to the EMDB with code EMD-27689.

p. Meprin alpha/fetuin-B structure has a residue in the disallowed Ramachandran region. It should be fixed.

This structure has already passed the validation criteria of the PDB. A single disallowed Ramachandran outlier is well within the acceptable tolerance for a model of this size and resolution. Despite exhaustive efforts, it is not always possible to satisfy all rotamer and Ramachandran outliers.

q. Table S1 their footnote needs to be improved. The Rosenthal and Henderson method should be cited. Their statement about sharpening “sometimes performed by deepEMhancer” is too vague. They need to be more specific about their methods.

We have cited the Rosenthal/Henderson paper as suggested. The statement about map sharpening has been adjusted to guide the reader to the appropriate maps which had been deposited, if they desire.

*“*As determined by the Rosenthal and Henderson method⁹³. For model building, amplitude corrected maps and deepEMhancer (which does not generate a B-factor) were used in combination. Where applicable, these maps have been deposited to the EMDB under the above codes.”*

Reviewer #2 (Remarks to the Author):

The authors adequately addressed all points raised, except for the requested extension of the aa sequence alignment for the interface patch shown in Fig. 2d. The authors state that they provide a structural alignment for >100 species, which would more precisely demonstrate the common structural fold. Indeed, from a structural point of view this is true, but the identified aa residues for ionic interaction within the oligomer, particularly around Arg372, are not displayed. The requested alignment is minor in silico work and should be presented in the supplementary material.

Reviewer #3 (Remarks to the Author):

All comments have been nicely addressed; i look forward to seeing this work being published.

Reviewer #4 (Remarks to the Author):

This revised manuscript addresses most of specific concerns raised by this reviewer. By reading the responses, the authors seem to address most of concerns by the other three reviewers as well. While the introduction is not improved, which, to the view of this reviewer, would affect the overall effectiveness to the readership of broader audiences that Nature communications intends to target, the authors have decided not to modify the introduction. I have only one concern, which is from the new data in supplemental figure 8b. It seems to show that meprin alpha wild type has a significant species of molecule that has the molecular weight lower than the expected while meprin alpha mutants and meprin beta do not have such problem. What is/are the reason for such species, e.g., proteolysis? Does such species affect the any major conclusion from the paper? The authors should comment on that. If it is the proteolysis issue that only affects wild type meprin alpha, one should resolve that.

REVIEWERS' COMMENTS

Reviewer #2 (Remarks to the Author):

The authors adequately addressed all points raised, except for the requested extension of the aa sequence alignment for the interface patch shown in Fig. 2d. The authors state that they provide a structural alignment for >100 species, which would more precisely demonstrate the common structural fold. Indeed, from a structural point of view this is true, but the identified aa residues for ionic interaction within the oligomer, particularly around Arg372, are not displayed. The requested alignment is minor in silico work and should be presented in the supplementary material.

We have included the raw multiple sequence alignment as a supplementary file which can be visualised in the reader's preferred visualisation software.

Owing to the discontinuous elements that make up the helical interface, the alignment of primary sequence is not very informative. We advise the reader to inspect the surface conservation (Figure 6) which is a 3D visual representation of the underlying multiple sequence alignment capturing >100 sequences.

Reviewer #3 (Remarks to the Author):

All comments have been nicely addressed; I look forward to seeing this work being published.

Reviewer #4 (Remarks to the Author):

This revised manuscript addresses most of specific concerns raised by this reviewer. By reading the responses, the authors seem to address most of concerns by the other three reviewers as well. While the introduction is not improved, which, to the view of this reviewer, would affect the overall effectiveness to the readership of broader audiences that Nature communications intends to target, the authors have decided not to modify the introduction. I have only one concern, which is from the new data in supplemental figure 8b. It seems to show that meprin alpha wild type has a significant species of molecule that has the molecular weight lower than the expected while meprin alpha mutants and meprin beta do not have such problem. What is/are the reason for such species, e.g., proteolysis? Does such species affect the any major conclusion from the paper? The authors should comment on that. If it is the proteolysis issue that only affects wild type meprin alpha, one should resolve that.

We do not believe this is from proteolysis, meprin α oligomers are very stable (no proteolysis on an SDS-PAGE after prolonged incubation). Rather, this is most likely a limitation of current mass photometry technology which is not capable of reliably measuring the mass of elongated large filaments. The observed mass fingerprint is skewed and includes shot noise. As stated in the figure legend (8b)

“The mass distribution of wild type meprin α appears skewed toward smaller oligomeric species, however this is artificial since the largest filaments are excluded during event detection. Nevertheless, the highly polydisperse nature of the wild-type sample is apparent.”

Specifically, the measurements of mass depend on reliable detection of contrast from single-molecule events which are then related to a standard curve (contrast-mass). Currently these events are detected by software and any events that do not produce a spherical point spread function are discarded. This is exactly the case for very large meprin oligomers which result in elongated point spread functions and long-range interference patterns. The apparent presence of “small” species likely originates from inaccuracies in event detection, overlapping interference from adjacent events, and shot noise. The shot noise is constant across all measurements, however it is more apparent in measurements of wild type meprin α than meprin β , since the signal-to-noise ratio is lower in the former (wild type meprin α events are discarded more frequently on average, while meprin β events readily rise above the shot noise).